# The function of ER-phagy receptors is regulated through phosphorylation-dependent ubiquitination pathways

Rayene Berkane [1,2], Hung Ho-Xuan [1,2,8], Marius Glogger [3,8], Pablo Sanz-Martinez[1,2,8], Lorène Brunello [1,2], Tristan Glaesner[2], Santosh Kumar Kuncha [1,2], Katharina Holzhüter [4], Sara Cano-Franco[1,2], Viviana Buonomo[5], Paloma Cabrerizo-Poveda [1,2], Ashwin Balakrishnan [3], Georg Tascher [1], Koraljka Husnjak [1], Thomas Juretschke[6], Mohit Misra[1,2], Alexis González [1], Volker Dötsch [4], Paolo Grumati [5,7], Mike Heilemann [3] & Alexandra Stolz [1,2] ✉

Selective autophagy of the endoplasmic reticulum (ER), known as ER-phagy, is an important regulator of ER remodeling and essential to maintain cellular homeostasis during environmental changes. We recently showed that members of the FAM134 family play a critical role during stress-induced ER-phagy. However, the mechanisms on how they are activated remain largely unknown. In this study, we analyze phosphorylation of FAM134 as a trigger of FAM134-driven ER-phagy upon mTOR (mechanistic target of rapamycin) inhibition. An unbiased screen of kinase inhibitors reveals CK2 to be essential for FAM134B- and FAM134C-driven ER-phagy after mTOR inhibition. Furthermore, we provide evidence that ER-phagy receptors are regulated by ubiquitination events and that treatment with E1 inhibitor suppresses Torin1-induced ER-phagy flux. Using super-resolution microscopy, we show that CK2 activity is essential for the formation of high-density FAM134B and FAM134C clusters. In addition, dense clustering of FAM134B and FAM134C requires phosphorylation-dependent ubiquitination of FAM134B and FAM134C. Treatment with the CK2 inhibitor SGC-CK2-1 or mutation of FAM134B and FAM134C phosphosites prevents ubiquitination of FAM134 proteins, formation of high-density clusters, as well as Torin1-induced ER-phagy flux. Therefore, we propose that CK2-dependent phosphorylation of ER-phagy receptors precedes ubiquitin-dependent activation of ER-phagy flux.

The endoplasmic reticulum (ER) is a continuum of membrane structures that dominate a considerable fraction of cell volume and form a central hub for many molecular events such as protein folding, lipid and steroid synthesis, as well as calcium storage[1]. The multifunctional nature of this organelle requires the protection of its integrity under multiple stress conditions to achieve optimal function of individual proteins and the entire organelle[2].

Selective turnover of the ER by autophagy (ER-phagy) has emerged as a major quality control pathway ensuring programmed renovation of the ER through elimination of discrete fragments

---

sequestered into autophagosomes and degraded through the lysosomal machinery (macroautophagy)[3]. Receptor-mediated macroautophagy (in the following called autophagy) relies on autophagy receptors selectively recognizing cargo, linking it to the autophagic membrane, and thereby promoting cargo sequestration in autophagosomes. To facilitate this function, most ER-phagy receptors are linked to the ER through a transmembrane domain (cargo binding) and also have an LC3 interaction region (LIR) that allows them to interact with members of the LC3/GABARAP (MAP1LC3 microtubule-associated proteins 1A/1B light chain 3 (LC3)/gamma aminobutyric acid receptor-associated protein) protein family on the autophagic membrane[4]. So far, eight membrane-embedded ER-phagy receptors have been identified: FAM134B/RETREG1[5], SEC62[6], RTN3L[7], CCPG1[8], ATL3[9], TEX264[10], FAM134A/RETREG2 and FAM134C/RETREG3[11], which participate in basal ER turnover, ER reshaping after stress-related expansion, as well as lysosomal degradation of selected ER proteins[2].

Members of the FAM134 family carry a conserved LIR motif located at their C-terminus, and their membrane domain consists of a reticulon homology domain (RHD), comprised of two hydrophobic regions embedded in the membrane that are bridged by a flexible cytoplasmic linker[12]. We recently showed that the three FAM134 proteins have shared and distinct functions and differ in the dynamics of their RHDs[11]. All FAM134 proteins provide critical functions in maintaining ER homeostasis and their individual absence results in massive deformation of the ER network and swelling of the ER. Although quite similar, the three paralogues may operate differently: while FAM134B is constitutively active, FAM134A and FAM134C predominantly exist in an inactive state under basal conditions[11]. Upon ER stress, all three receptors can induce significant fragmentation and degradation of the ER via the lysosome. We also demonstrated that FAM134C can act in concert with FAM134B and enhances its activity in degrading misfolded procollagen[11]. The cooperative performance of these two paralogues is likely to be tightly regulated through processes including post-translational modifications (PTMs) and structural organization. Along these lines, it was recently shown that phosphorylation of FAM134B-RHD, within the flexible cytoplasmic linker by CAMKII (Ca$^{2+}$/calmodulin-dependent protein kinase II), drives the oligomerization of FAM134B and promotes ER fragmentation prior to ER-phagy[13]. The question of how phosphorylation mechanistically drives oligomerization and if other kinases are involved in this regulatory pathway remains unknown.

Here, we show that the activity of both FAM134B and FAM134C can be modulated by inhibiting mTOR (the master kinase of autophagy) with Torin1 and identify several downstream kinases using a chemical screening approach. In this unbiased screen, we identify CK2 downstream of Torin1-induced and FAM134B-, as well as FAM134C-driven ER-phagy. Phosphorylation sites within the RHD of FAM134 proteins were important for ER-phagy induction by Torin1 treatment. Phosphorylation of FAM134B and FAM134C was also a prerequisite for subsequent ubiquitination of FAM134B and FAM134C and the formation of high-density receptor clusters. This suggests phosphorylation-dependent ubiquitination of both FAM134 proteins as the driving force behind high-density clustering of FAM134B and FAM134C at active ER-phagy sites.

## Results

### ER-phagy driven by FAM134C and FAM134B is activated upon inhibition of mTOR

Our recent findings showed that FAM134C is an ER-phagy receptor that works in concert with FAM134B and that overexpressed FAM134C is inactive under basal conditions and activated upon starvation[11]. We hypothesized that phosphorylation events activate ER-phagy and tested the impact of mTOR inhibition on ER-phagy flux driven by FAM134C and FAM134B using the ER-phagy reporter ssRFP-GFP-KDEL[10] (later referred to as the KDEL reporter). GFP is rapidly quenched within the acidic environment of the lysosomes, while the fluorescence of RFP is preserved. Therefore, changes in the RFP/GFP ratio (total integrated

fluorescence) can be used to compare the ER-phagy flux between conditions independent of the monitored cell number (Fig. 1A). Several ER-phagy reporters have been identified so far[5–11] and activation of any of these receptors can, in principle, affect KDEL monitored flux. To specifically measure FAM134-driven ER-phagy flux and minimize the impact of other ER-phagy receptors, we used U2OS cell lines constitutively expressing the KDEL reporter (CTRL) and stably overexpressing HA-FLAG-FAM134B (FAM134B) and HA-FLAG-FAM134C (FAM134C) in a doxycycline (DOX)-inducible manner[11]. Under basal conditions (DMSO), overexpression of FAM134B or FAM134C slightly increased ER-phagy flux compared to control cells (Fig. 1B). Upon treatment with 250 nM Torin1, the flux of ER-phagy increased strongly over time in all cell lines tested and specifically upon overexpression of FAM134B or FAM134C. Induction of ER-phagy flux was ~3-fold higher in FAM134B and FAM134C overexpressing cells compared to CTRL cells, indicating that under these conditions the vast majority of measured ER-phagy flux results from FAM134-driven ER-phagy (Fig. 1B). We then investigated whether loss of endogenous Fam134b or Fam134c decreases the amount of Torin1-induced ER-phagy (Fig. 1C, D). For that, we used mouse embryonic fibroblasts (MEFs) isolated from *Fam134b* and *Fam134c* single knockout (KO) mice complemented with the ssRFP-GFP-KDEL reporter[11]. Our data confirms that the contribution of FAM134 proteins to mTOR-mediated ER-phagy is an endogenous function of FAM134. The difference between ER-phagy flux in cells overexpressing FAM134B or FAM134C and the CTRL cell line represents Torin1-mediated, FAM134-driven ER-phagy flux and can serve as a screening window to identify factors important for FAM134 activation (Fig. 1B). Notably, the mTORC1 inhibitor Rapamycin has a similar, but weaker impact on ER-phagy flux (Supplementary Fig. 1A). All together, these findings suggest that FAM134s function is regulated through mTOR inhibition.

### Kinases involved in mTOR-mediated and FAM134-driven ER-phagy

To understand which downstream kinases regulate FAM134B and FAM134C-driven ER-phagy upon mTOR inhibition, we screened a library of ~100 highly selective kinase inhibitors (chemogenomic set of EUbOPEN; see EUbOPEN.org) and available negative controls for their potency to prevent ER-phagy induction. These target-optimized inhibitors have a $K_i$ in the nanomolar range and the screening was performed uniformly with 1 μM final compound concentration. After 24 h of DOX treatment to induce FAM134B and FAM134C overexpression, respectively, cells were preincubated for 1 h with kinase inhibitors to block target kinase activity (−1 h) and subsequently treated with 250 nM Torin1 to induce ER-phagy (0 h). Time-point measurements of the ratio of total integrated red (RFP) and green (GFP) fluorescence, as well as cell confluence (phase), were taken every 2 h over a total period of 48 h. Heatmaps summarizing the time-resolved screening results on ER-phagy flux and cell growth are shown in Fig. 1E and Supplementary Fig. 1B, respectively. Inhibition of several kinases prevented Torin1-induced, FAM134-driven ER-phagy, indicated by the loss of red color toward higher time points in the heatmap (Fig. 1E). To present the screening results in a visually receptive format, the data were transformed and normalized: the difference in the RFP/GFP ratio between the untreated and Torin1-treated cells was defined as 100% (Supplementary Fig. 1C). Early time points (<12 h) were excluded due to the small screening window and subsequent misinterpretation of the early data points. The impact of individual compounds was represented as % of the relative reduction of FAM134-mediated ER-phagy per time point (Supplementary Fig. 1C). Consequently, ER-phagy relevant compounds (HITs) were defined as compounds causing >20% relative reduction in ER-phagy flux over a minimum time period of 12 h (6 time points). We identified 10 kinase inhibitors with impact on FAM134B and 15 kinase inhibitors with impact on FAM134C activation, with 8 overlapping HITs (Supplementary Figs. 1D and 2A). From identified HITs, we were

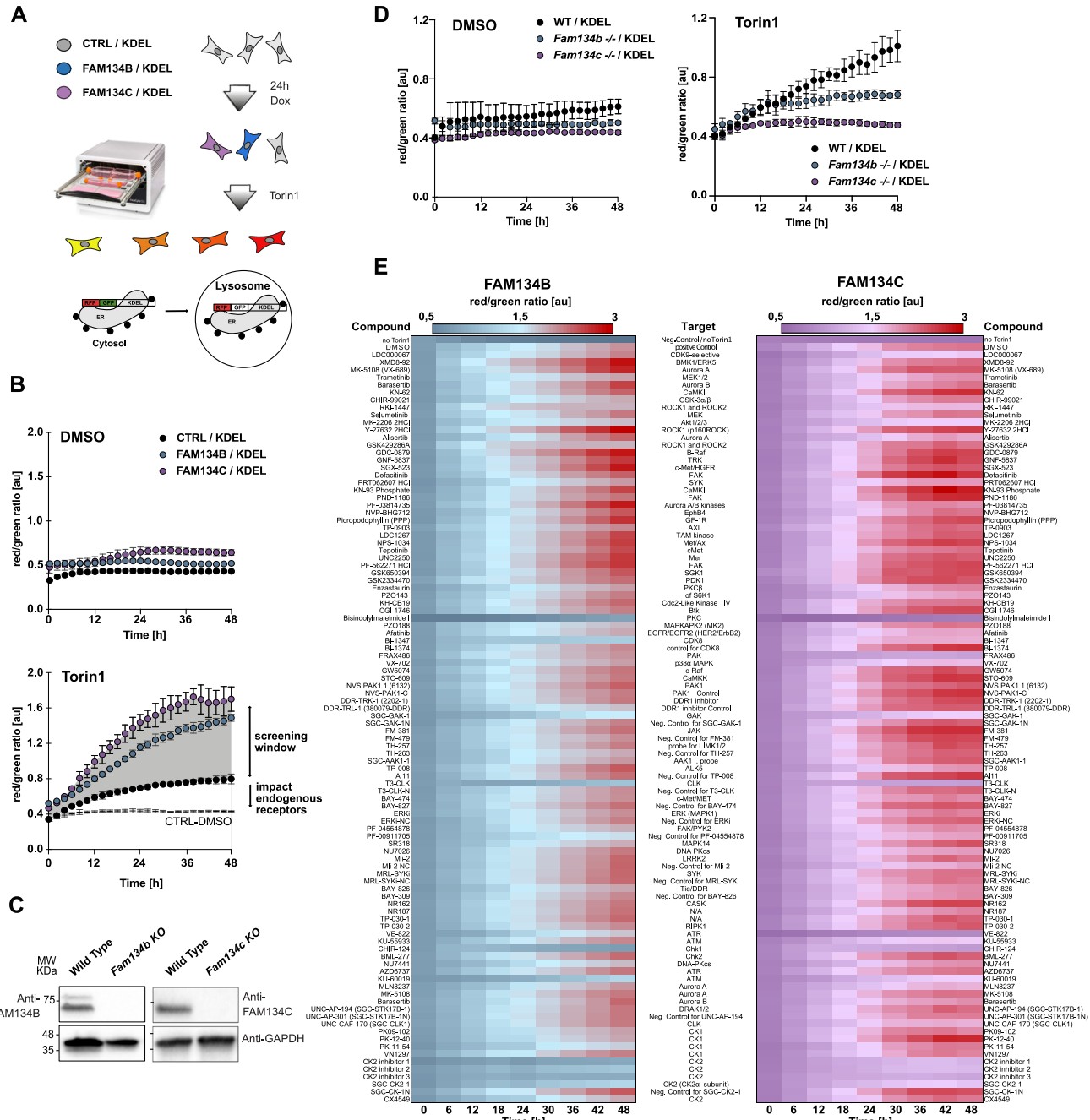

**Fig. 1 | Chemical screen to identify kinases involved FAM134B- and FAM134C-driven ER-phagy. A** Schematic representation of the ER-phagy flux assay using U2OS cells constitutively expressing the ssRFP-GFP-KDEL reporter and in addition overexpressing FAM134B (FAM134B) or FAM134C (FAM134C) in DOX-inducible manner. Cells carrying the empty overexpression vector are used as control cells (CTRL). ER-phagy flux is represented by the red/green (RFP/GFP) ratio of the total integrated fluorescence intensities. Higher values indicate higher ER-phagy flux. Figure created with Affinity Designer by Serif. **B** ER-phagy flux analysis under basal (DMSO) and Torin1-induced conditions in the absence and presence of FAM134B/C overexpression. Differences in the ER-phagy flux of CTRL and FAM134B/C over-expressing cells (indicated as gray shade) serve as the screening window for (**E**) to identify kinases involved in FAM134B/C-driven ER-phagy upon Torin1 treatment. **C** Representative blots image of Fam134 proteins in indicated MEFs isolated from

*Fam134b* and *Fam134c* KO mice, expressing ER-phagy reporter ssRFP-GFP-KDEL. $n = 3$ biological replicates. **D** ER-phagy flux in ssRFP-GFP-KDEL *Fam134* WT and KO MEFs, represented by the RFP/GFP ratio of total integrated intensities in basal (DMSO) or induced (Torin1) conditions. **E** Heatmap of an ER-phagy flux screen of ~100 probe and probe-like kinase inhibitors and available negative control compounds (1 μM final concentration) in the presence of 250 nM Torin1, using U2OS cells described in (**A**). Total integrated fluorescent intensities (RFP/red and GFP/green) were monitored in the IncuCyte® S3 over a time course of 48 h. Data points represent the averaged red/green ratio of three images comprising >100 cells and taken from $n = 3$ independent wells. Data information: data shown in (**B**)−(**E**) represent averaged data obtained from $n = 3$ individual wells via the IncuCyte® S3, each view containing >100 cells. Data are mean ± SD, [au] arbitrary unit. Source data are provided as source data file.

specifically interested in those that target kinases closely related to known mTOR signaling pathways and not inducing obvious cell death. Five candidate compounds corresponded to these criteria: VE-822, KU-60019, CHIR-124, MK-2206 2HCL and SGC-CK2-1 targeting ATR, ATM, Chk1, Akt, and CK2, respectively (Supplementary Fig. 2A, B). These kinases were reported to be related to or part of the mTOR signaling axis. For instance, ATM and ATR, which orchestrate the DNA damage response (DDR), are members of the phosphatidylinositol 3-kinase (PI3K)-related family (PI3KKs) that also comprises mTOR[13,14]. CK2 was previously associated with mTOR, as its inhibition was shown to have an anticancer effect through downregulation of the AKT/mTOR signaling pathway[15]. For validation, selected HITs were retested for dose-response. Five selected HIT compounds, VE-822, KU-60019, CHIR-124, MK-2206 2HCL and SGC-CK2-1 were confirmed and blocked FAM134B, as well as FAM134C-mediated ER-phagy in a dose-dependent manner (Fig. 2A, B). We also tested the impact of selected HIT compounds on Torin1-induced ER-phagy in U2OS CTRL cells. Here, the screening window is relatively narrow. However, HIT compounds prevented Torin1-induced ER-phagy to some extent, indicating endogenous regulatory function of the target kinases (Supplementary Fig. 2C).

### mTOR inhibition triggers the phosphorylation of FAM134B and FAM134C by CK2

Parallel to the phenotypic chemical screening, the phosphorylation status of FAM134B and FAM134C was analyzed using a mass spectrometry (MS) approach (Fig. 3A, B). Several phosphorylation sites on highly conserved residues were identified under basal or Torin1-induced conditions (Fig. 3B and Supplementary Fig. 3A) including previously reported functional sites within FAM134B[16]. While this in principle supports the robustness of our MS approach, we were surprised that an inhibitor of the reported CAMKII kinase was absent from the list of our HIT compounds (Fig. 1E and Supplementary Fig. 2A). The previous report tested KN-93, a CaMKII inhibitor with a $K_i$ of 2.58 μM at 10 μM[16]. Although KN-93 is relatively selective for CaMKII, it has been found to have several off-target effects at higher concentrations that can indirectly affect FAM134-dependent ER-phagy flux, including effects on $Ca^{2+}$ channels, $Ca^{2+}$ release, and calmodulin activity[17]. Our screen included a more potent inhibitor of the CaMKII kinase with a $K_i$ of 0.37 μM (KN-93 phosphate). We reanalyzed both inhibitors in dose-response and found doses of >4 μM KN-93 to impact ER-phagy flux over time. KN-93 phosphate showed no effect (defined as >20% relative reduction over min. 12 h) up to 2 μM (Supplementary Fig. 2D).

Identified phosphorylation sites were also analyzed using the motif-match analysis tool PHOSIDA[18] to predict kinases. Casein kinase 2 (CK2) was the most frequently predicted kinase for our data set, suggesting that CK2 can directly phosphorylate FAM134 proteins (Supplementary Fig. 3B). The prediction nicely correlated with our screening results, where the CK2 inhibitor SGC-CK2-1 reduced up to 60% and 70% of Torin1-induced ER-phagy in FAM134B and FAM134C overexpressing cells, respectively (Fig. 2A, B). To further validate our hypothesis that CK2 directly phosphorylates FAM134 proteins, we analyzed the ability of purified CK2 to phosphorylate FAM134B in vitro. A reaction of FAM134B and CK2 (subunits CK2A1 and CK2A2) purified from bacteria showed a shift of FAM134B on a phos-tag gel, indicating CK2-dependent phosphorylation (Fig. 3C). To validate the in cellulo phosphorylation sites within the RHD domain of FAM134B, the assay was repeated with purified FAM134B-RHD and analyzed by MS (Supplementary Fig. 3C). We could identify one of the potentially three phosphorylation sites in vitro, indicating that CK2 is indeed capable of directly phosphorylating FAM134B (Fig. 3D, E and Supplementary Fig. 3D).

### CK2 activity positively impacts FAM134-driven ER-phagy

For validation of the functional impact of the identified phosphorylation sites, stable inducible cell lines expressing mutated FAM134B S149,151,153A or FAM134C S258A (phospho-mutants) were generated

and subjected to our ER-phagy flux assay (Fig. 3E, F). The FAM134C S258A mutation did not impact ER-phagy flux under basal conditions, however, had a strong negative impact on Torin1-induced ER-phagy flux. Mutation of the FAM134B phosphorylation sites also negatively affected the ability of FAM134B to increase the ER-phagy flux after Torin1-induction. At the same time, the FAM134B phospho-mutant increased basal ER-phagy flux—visible by a slightly higher steady-state red/green ratio at 0 h—indicating stress/a dominant negative effect upon expression of the mutant protein. In line with these data, loss of endogenous CK2 prevented induction of ER-phagy flux after Torin1 treatment in cells overexpressing FAM134B and FAM134C, respectively (Supplementary Fig. 3E, F).

We previously showed that FAM134 proteins are involved in the degradation of misfolded collagen[11]. To test our hypothesis that phosphorylation of FAM134 proteins is necessary for the activation of receptors and the maintenance of a high ER-phagy flux capacity in times of stress, we reconstituted MEFs lacking Fam134b (*Fam134b* KO) and Fam134c (*Fam134c* KO), with respective phospho-mutants. In contrast to the wild-type (WT) protein, FAM134 proteins lacking identified phosphosites were unable to recover the endogenous function of the WT protein and showed elevated levels of collagen (Fig. 3G–J). Our findings stress the importance of the identified phosphorylation sites in the activation process of FAM134 proteins.

### Phosphorylation by CK2 triggers ubiquitination of FAM134

When immunoprecipitating FAM134C for MS experiments, we noticed dominant bands detected by anti-HA-antibody (FAM134C) > 50 kDa on control western blots. This indicated PTMs beyond phosphorylation that can shift the protein to higher molecular weights. One possibility was multi- or poly-ubiquitination. To test this hypothesis, we used the tandem ubiquitin binding entity 2 (TUBE 2) system, which is capable of pulling down mono- and poly-ubiquitinated proteins (Supplementary Fig. 4A). Using lysates of cells overexpressing HA-FAM134B and HA-FAM134C, we were able to identify ubiquitinated species of FAM134B and FAM134C, respectively (Supplementary Fig. 4B). The amount of ubiquitinated FAM134 species increased after treatment with Bafilomycin A1 (BafA1), indicating that ubiquitinated FAM134 proteins indeed undergo ER-phagy and therefore accumulate after blocking lysosomal degradation (Supplementary Fig. 4B).

To test whether ubiquitination is functionally important for FAM134-driven ER-phagy, we used our ER-phagy flux assay and co-treated cells with Torin1 and different concentrations of an E1 inhibitor (MLN7243) blocking the first step of ubiquitination. Inhibition of the ubiquitin system had a negative impact on ER-phagy flux, and higher concentrations of the E1 inhibitor suppressed FAM134-driven ER-phagy to more than 80% (Fig. 4A, B). Along the hypothesis that FAM134 ubiquitination promotes ER-phagy, a higher amount of ubiquitinated species was found for endogenous FAM134B and FAM134C (Fig. 4C), as well as in the overexpression system (Fig. 4D, E) upon Torin1-induction. With a ~3.4-fold increase, the difference in ubiquitination levels between basal and induced conditions was much more pronounced for FAM134B than for FAM134C (~1.4-fold) (Fig. 4E).

We then asked the question of whether there is a cross-talk between phosphorylation and ubiquitination. Therefore, the ubiquitination status of FAM134B and FAM134C after Torin1 treatment was analyzed in the presence and absence of the CK2 inhibitor. High-molecular species disappeared after co-treatment with Torin1 and CK2 inhibitor or co-treatment with the E1 inhibitor (control) and CK2 inhibitor, indicating that CK2-mediated phosphorylation is indeed a prerequisite for the ubiquitination of FAM134 proteins (Fig. 4C–E). Likewise, genetic depletion of CK2 prevented ubiquitination of FAM134 proteins under Torin1-induced conditions (Fig. 4F, G). To exclude the possibility that other CK2-mediated phosphorylation

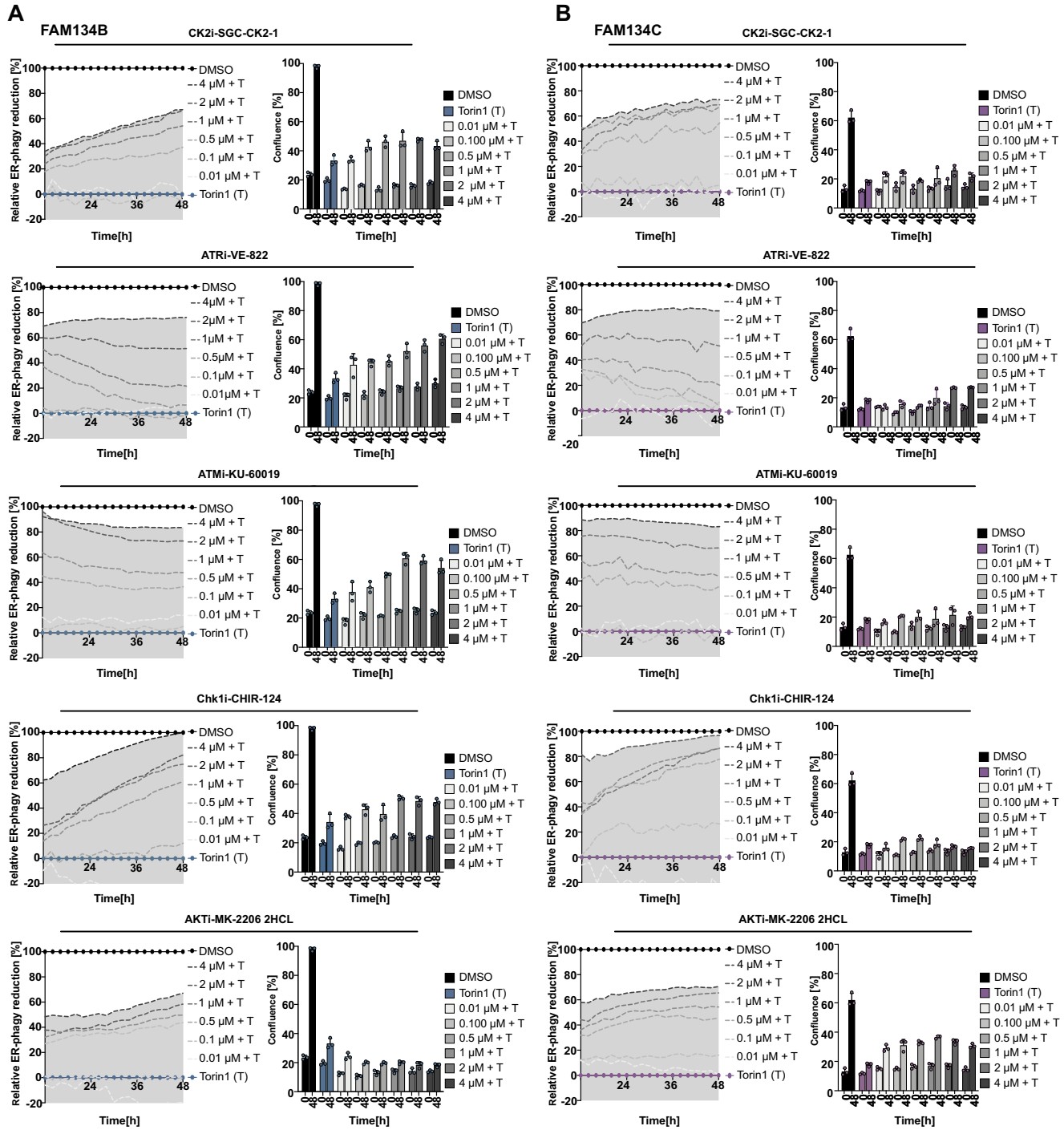

**Fig. 2 | mTOR inhibition triggers phosphorylation of FAM134B and FAM134C by CK2.** Validation of kinase screening HITs counteracting the effect of Torin1 in FAM134B- (**A**) or FAM134C- (**B**) overexpressing cells. Left panels: analytic representation of FAM134B or FAM134C-induced ER-phagy changes upon treatment with a dilution series of selected compounds (SGC-CK2-1 (CK2i), VE-822 (ATRi), KU-60019/KU-55933 (ATMi), CHIR-124 (Chk1i) and MK-2206 2HCL (AKTi)) ranging from 0.01 to 4 μM combined with Torin1 (T) treatment. DMSO was set to represent 100% inhibition of FAM134B- and FAM134C-induced ER-phagy while Torin1 represents 0%. Averaged RFP/GFP ratio measured in response to HIT compound treatments was normalized at each time point to the averaged RFP/GFP ratio of DMSO and Torin1. The area under the highest concentration curve was colored in gray for better visualization. Right panels: growth of U2OS cells overexpressing FAM134B or FAM134C treated with indicated compounds and concentrations and recorded over a time period of 48 h (data from two time points are shown in the bar graphs). Data information: data shown in (**A**, **B**) represent averaged data obtained from *n* = 3 individual wells via the IncuCyte® S3, each view containing >100 cells. Data are mean ± SD, [au] arbitrary unit. Source data are provided as source data file.

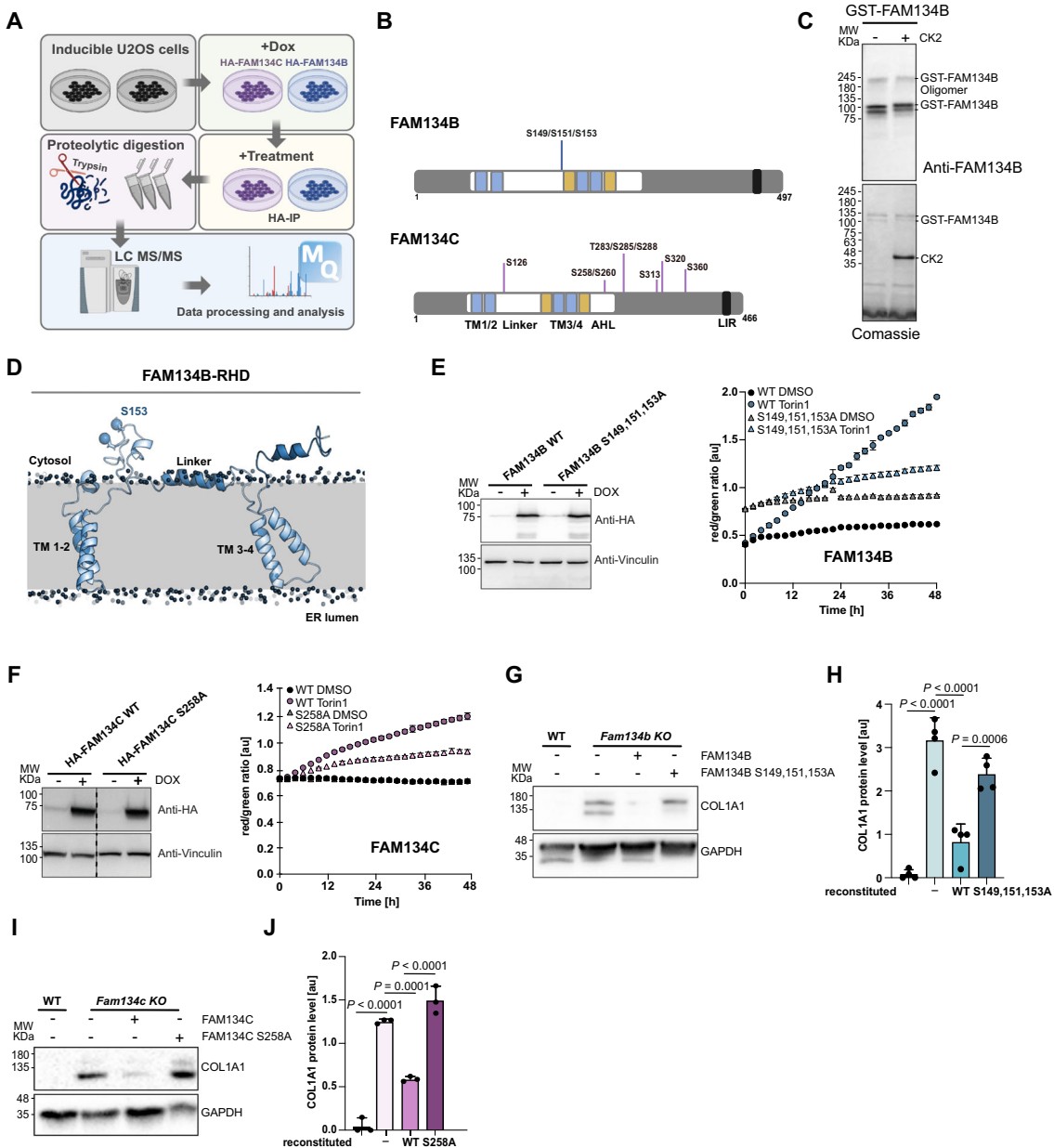

**Fig. 3 | CK2 activity positively impacts FAM134-driven ER-phagy. A** Schematic representation of the mass spectrometry experimental procedure to identify FAM134s phosphorylation sites under basal (DMSO) or induced (Torin1) conditions. Created with BioRender.com. **B** Schematic representation of FAM134B or FAM134C proteins highlighting the reticulon homology domain (RHD; white) comprising two transmembrane domains (TM1/2 and TM3/4; blue) flanked by membrane associated linker regions (green), the conserved LC3 interacting motif (LIR), as well as sites of phosphorylation events. **C** Representative blot image of in vitro phosphorylation of full-length GST-FAM134B by purified CK2 and corresponding Coomassie-stained gel (lower panel). Anti-GST antibody was used to detect GST-FAM134B (upper panel). $n = 3$ biological replicates. **D** Schematic three-dimensional representation of FAM134B-RHD highlighting serine 153 phosphorylated in vitro by CK2. **E**, **F** Left panels: representative blot images validating the stable expression of FAM134 WT or phospho-mutant proteins under DOX induction in the background of U2OS cells expressing ER-phagy reporter ssRFP-GFP-

KDEL. Right panels: ER-phagy flux in U2OS cells expressing FAM134s WT or phospho-mutants and ER-phagy reporter ssRFP-GFP-KDEL, represented by the RFP/GFP ratio of total integrated intensities in basal (DMSO) or induced (Torin1) conditions. These data represent averaged data obtained from $n = 3$ individual wells via the IncuCyte® S3, each view containing >100 cells. Data are mean ± SD, [au] arbitrary unit. Representative blot image (**G**) and the relative bar plot (**H**) showing Collagen I accumulation in WT MEFs, Fam134b KO or cells reconstituted with FAM134B WT and phospho-mutant (S149,151,153A). Representative blot image (**I**) and the relative bar plot (**J**) showing Collagen I accumulation in WT MEFs, Fam134c KO or cells reconstituted with FAM134C WT and phospho-mutant (S258A). Data information: GAPDH was used as a reference for ratio calculation. The statistical significance was estimated by one-way ANOVA. $n = 4$ biological replicates for (**G**, **H**) and $n = 3$ biological replicates for (**I**, **J**). Data are mean ± SD, [au] arbitrary unit. Source data are provided as source data file.

events are the cause of ubiquitination of FAM134, we also analyzed the ubiquitination status of our FAM134 phospho-mutants. Indeed, FAM134B and FAM134C phospho-mutants had no ubiquitin modifications both under basal and Torin1-induced conditions (Supplementary Fig. 4C).

## Oligomeric FAM134B and FAM134C is ubiquitinated in a CK2-dependent manner

Receptor oligomerization is believed to be important for vesiculation/fragmentation of the ER and subsequently delivery to the lysosome[12,19]. Therefore, we investigated the ubiquitination status of oligomerized

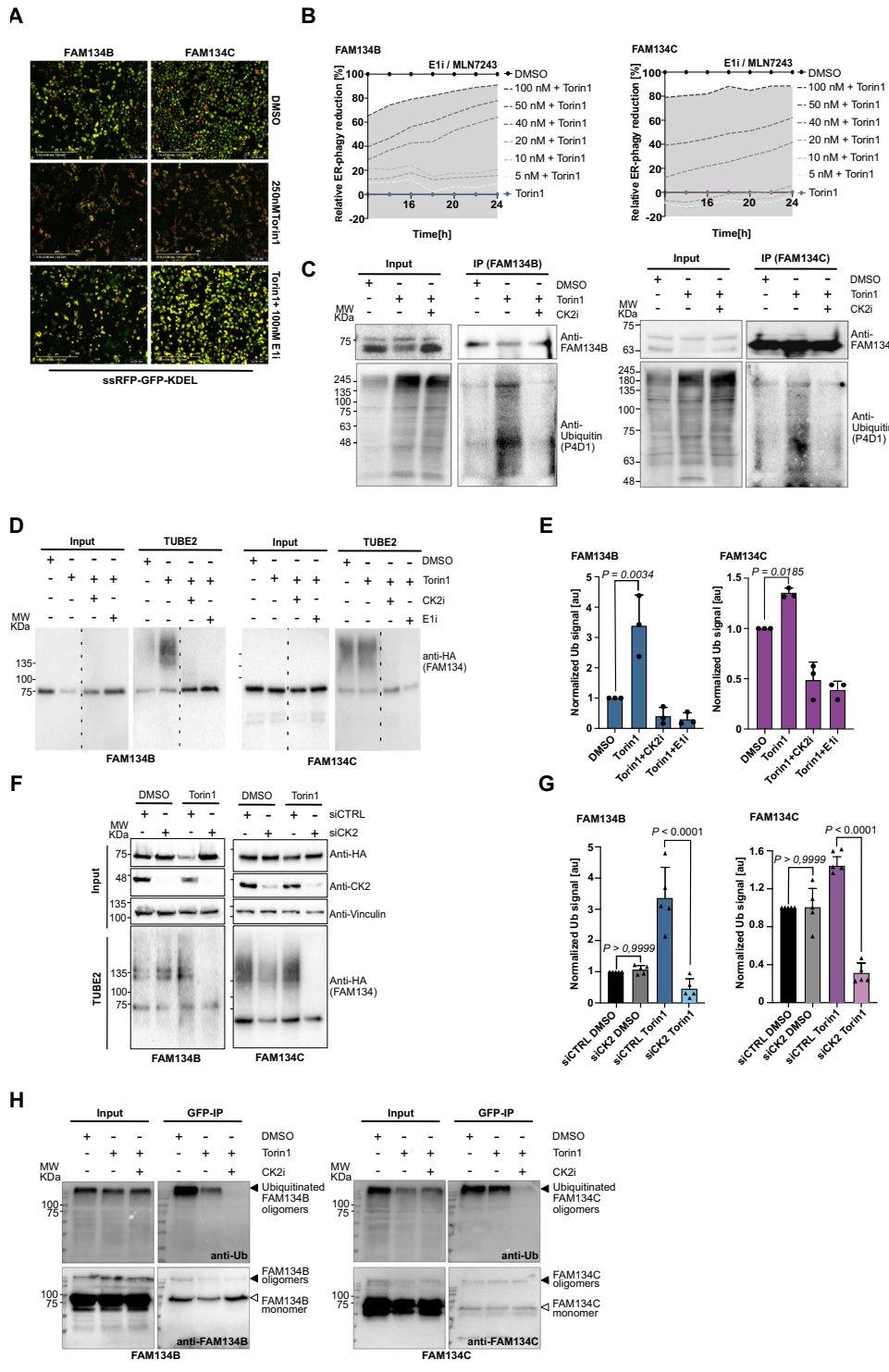

FAM134 proteins by performing immunoprecipitation experiments using the split Venus system[20], which drives and preserves dimerization (Supplementary Fig. 4D). Two constructs carrying half of the GFP were generated for FAM134B and FAM134C, respectively. After transient co-transfection of both constructs, FAM134 proteins oligomerize and split GFP molecules can fuse to form a complete and stable GFP molecule (Supplementary Fig. 4D, E). We immunoprecipitated FAM134B or FAM134C homo-dimers using GPF-Trap beads and

detected for ubiquitin. Again, we were able to detect high-molecular weight species of FAM134B and FAM134C under basal and induced conditions (Fig. 4H). These may represent SDS-resistant dimers as suggested previously[16], as well as multi/poly-ubiquitinated FAM134 proteins. The detected ubiquitination signal completely vanished upon co-treatment with the CK2 inhibitor, further supporting our claim that CK2-dependent phosphorylation events are a prerequisite for the ubiquitination of FAM134B and FAM134C.

**Fig. 4 | FAM134B and FAM134C phosphorylation by CK2 is prerequisite for their ubiquitination and Torin1-dependant ER-phagy induction. A** Exemplary pictures of U2OS cells overexpressing FAM134B or FAM134C and stably expressing ssRFP-GFP-KDEL reporter and exposed to indicated conditions. Pictures were acquired using the IncuCyte® S3[10x] and show the overlaid GFP and RFP signal from DMSO, Torin1 and Torin1 + E1 inhibitor treated cells at time point 24 h. **B** Quantitative representation of ER-phagy flux essay performed in (**A**) analyzing changes over time in FAM134B- (left panel) and FAM134C-induced ER-phagy (right panel) upon treatment with E1 inhibitor dilution series in combination with 250 nM Torin1. Data shown in (**A, B**) represent averaged data obtained from $n = 3$ individual wells via the IncuCyte® S3, each view containing >100 cells. Data are mean ± SD, [au] arbitrary unit. **C** Representative blot image of endogenously immunoprecipitated FAM134 proteins from total cellular extracts collected from U2OS cells previously treated for 6 h with the indicated treatments. Anti-FAM134B or anti-FAM134C antibodies were used to detect endogenous proteins while anti-Ub (P4D1) was used to detect ubiquitination. $n = 3$ biological replicates. Representative blot image (**D**) and corresponding quantification (**E**) of TUBE 2 assay in U2OS cells overexpressing FAM134B (left panel) or FAM134C (right panel) subjected for 6 h to the indicated conditions prior to cell lysis. anti-HA antibody was used to detect HA-tagged FAM134B and FAM134C and ubiquitinated species. Representative blot image (**F**) and relative bar plot (**G**) of TUBE 2 assay in U2OS cells overexpressing FAM134B (left panel) or FAM134C (right panel) transfected with control siRNA (siCTRL) or siPOOL targeting CSNK2A1 and CSNK2A3 (siCK2). After 48 h siRNA transfection, cells were DOX-induced overnight and treated for 6 h with indicated conditions prior to cell lysis. anti-HA antibody was used to detect HA-tagged FAM134B and FAM134C and ubiquitinated species while anti-CK2 antibody was used to detect endogenous CK2 levels. **H** Representative western blot of the bimolecular complementation affinity purification assay (BiCAP) showing ubiquitination status of FAM134B (left panel) and FAM134C (right panel) dimers. GFP-Trap pulldown was performed to immunoprecipitate reconstituted GFP representing FAM134B or FAM134C dimers. Anti-FAM134B or anti-FAM134C antibodies were used to detect endogenous proteins, while anti-Ub (P4D1) was used to detect ubiquitination. $n = 3$ biological replicates. Data information: for TUBE 2 assays, DMSO condition was used as reference for ratio calculation. The statistical significance was estimated by one-way ANOVA. $n = 3$ biological replicates for (**D, E**) and $n = 5$ biological replicates for (**F, G**). Data are mean ± SD, [au] arbitrary unit. Source data are provided as source data file.

## Phosphorylation increases the size of the FAM134B nanoscale cluster

To refine the impact of FAM134-RHD phosphorylation on clustering we used single-molecule localization microscopy (SMLM), an optical super-resolution method that achieves near-molecular resolution[21] and enables a quantitative readout[22] to analyze nanoscale clustering (50–200 nm) of FAM134 proteins. Optimized sample preparation conditions consisted of mild HA-FAM134 overexpression (12 h of DOX induction), followed by 6 h of treatment with DMSO, Torin1, and Torin1 + CK2 inhibitor, respectively. Using antibodies targeting the HA epitope in combination with accumulation of DNA points in nanoscale topography (DNA-PAINT)[22], we visualized FAM134 clusters with sub-diffraction spatial resolution (Fig. 5A, B). A quantitative analysis of FAM134B nanoscale clustering (see "Methods") revealed a unimodal distribution of cluster diameters under basal conditions (mode = 66 nm, $\sigma = 0.23$ nm) (Fig. 5C). Upon Torin1 treatment, a second population emerged at a larger cluster diameter (mode = 104 nm, $\sigma = 0.24$ nm), while the first population was maintained. Cotreatment with the CK2 inhibitor prevented the formation of the larger nanoscale cluster population, while retaining the first population of small clusters (mode = 69 nm, $\sigma = 0.23$ nm) (Fig. 5C), indicating that the size of FAM134B clusters is mediated and positively regulated by CK2 activity. A quantitative analysis of the diameters of FAM134C nanoscale clusters also showed a unimodal, but broader distribution under basal conditions (mode = 52 nm, $\sigma = 0.52$ nm) (Fig. 5D). Unlike FAM134B, Torin1 treatment did not lead to the emergence of a second population of FAM134C clusters with larger diameter (mode = 53 nm, $\sigma = 0.51$ nm) (Fig. 5D). However, statistical analysis revealed a small, yet significant change in the diameter of the overall cluster, which was reversed by co-treatment with the CK2 inhibitor (Supplementary Fig. 5A). Because of the relatively large size of the HA-tag and HA-antibodies that may hinder efficient labeling of dense protein assemblies, we additionally generated a stable inducible cell lines expressing FAM134 tagged with the recently developed ALFA-tag (Supplementary Fig. 5D)[23]. Speed optimized DNA-PAINT microscopy[24] confirmed our findings that Torin1 treatment increased the average size of FAM134B clusters, while the size of the FAM134C cluster remained unchanged (Supplementary Fig. 5E). Using shorter induction times, we observed a unimodal population of FAM134B cluster diameters for Torin1-treated cells under these settings (Supplementary Fig. 5E), indicating that the majority of visualized clusters change from an inactive to an active ER-phagy site. In addition, these experiments revealed that inhibition of CK2 not only reverts the size of the FAM134B cluster, but also caused a significant decrease compared to the basal conditions (DMSO) (Supplementary Fig. 5E). This presumably represents the small contribution of endogenous ER-phagy receptors to the visualized clusters, which is only detected by the advanced ALFA-tag system. The impact of inhibition of CK2 on the diameters of the FAM134B cluster can be caused by direct (phosphorylation of FAM134B) or indirect actions (phosphorylation of other regulatory or functional proteins within the ER-phagy process). To support a direct regulation of FAM134 by CK2, we analyzed the established phospho-mutants of FAM134B and FAM134C. The cluster diameter distribution of the phospho-mutant of FAM134B was similar to that of FAM134B WT under basal conditions (mode = 63 nm, $\sigma = 0.17$ nm), however, did not increase after Torin1 treatment (mode = 65 nm, $\sigma = 0.19$ nm) (Fig. 5C). This indicates that mutated phosphorylation sites have a direct impact on the organization of FAM134B and that CK2-mediated phosphorylation drives/boosts the formation of larger FAM134B nanoscale clusters after ER-phagy induction by Torin1. Regarding the wild-type form of FAM134C, we did not observe an impact of inhibition of CK2 on the diameters of the FAM134C S258A cluster diameters (Fig. 5D). Our data therefore suggests that serine 258 phosphorylation has regulatory functions that are not impacting the size of FAM134C clusters in our overexpression system.

## CK2-mediated phosphorylation is essential for the formation of high-density clusters of FAM134B and FAM134C

The number of FAM134 proteins within nanoscale clusters may vary between clusters of the same size, depending on the space between individual FAM134 molecules (Figs. 5A and 6A). Close packing of FAM134 proteins (high density) may be required for their ER-phagy function, e.g., bending of the ER membrane to facilitate efficient fragmentation[12]. To further refine the characteristics of nanoscale clusters, we analyzed the relative protein density within individual FAM134 nanoclusters by extracting the number of single-molecule binding events (localizations) for different conditions and phosphorylation states (phospho-mutants) (Fig. 5E–J). A direct comparison of the WT and FAM134 phospho-mutants showed that clusters with high numbers of localizations (=high density) were largely absent for the inactive phospho-mutants (colored dots versus colored triangles). We then defined wild-type high-density FAM134 clusters by thresholding (>100 localizations/cluster, cluster diameter >90 nm; gray regions). For FAM134B, the number of high-density clusters (as percentage of all analyzed clusters) increased after Torin1 treatment (5.6%) compared to baseline conditions (2%) and inhibition of CK2 (3.7%) (Fig. 5E–G). Interestingly, we measured only a minor increase in high-density clusters for FAM134C after Torin1 treatment (6.6% for DMSO and 7.3%) but a strong decline after CK2 inhibition (2.6%) (Fig. 5H–J). This correlates with the ubiquitination status of FAM134C and indicates that FAM134C is regulated by CK2 also under

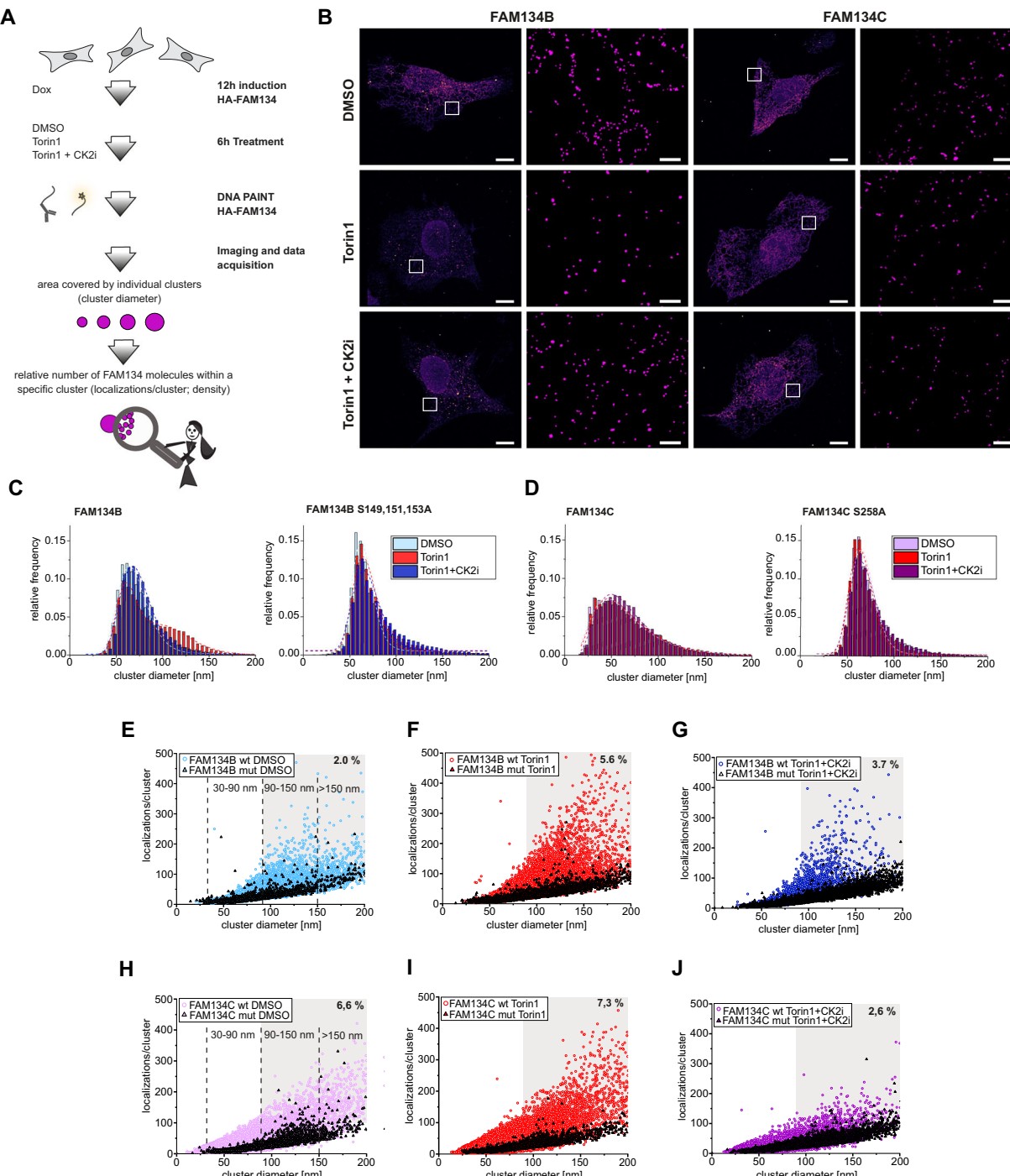

**Fig. 5 | Phosphorylation of FAM134B and FAM134C proteins impacts their clustering. A** Schematic representation of sample preparation for exchange DNA-PAINT experiment performed on U2OS overexpressing FAM134B and FAM134C upon 12 h DOX induction. Cells were grown under basal, 250 nM Torin1 or 250 nM Torin1 + 1 μM CK2 inhibitor conditions. Figure created with Affinity Designer by Serif. **B** Super-resolved DNA-PAINT images showing HA-FAM134B (left panel) and HA-FAM134C (right panel) labeling in U2OS cells treated as described in (**A**). Localization of single molecules and reconstruction of clusters are annotated in the region of interest. Scale bars: 10 and 1 μm. Quantitative analysis of FAM134B (**C**) and FAM134C (**D**) nanocluster diameters in indicated conditions (250 nM Torin1 or 250 nM Torin1 + 1 μM CK2 inhibitor). Quantification was performed via DBSCAN clustering algorithm 3 and distribution of clusters was assessed in WT FAM134B- and FAM134C-overexpressing U2OS cells and their respective phospho-mutants (FAM134B S149,151,153A and FAM134C S258A) (**E–J**). Comparative analysis of the relative protein density for FAM134B or mutant (**E–G**) and FAM134C or mutant (**H–J**). Frequency of transient binding events (localization) per nanoscale cluster was analyzed under basal, 250 nM Torin1 or 250 nM Torin1 + 1 μM CK2 inhibitor conditions. Data information: super-resolution microscopy experiments have been repeated four times with consistent, similar results. An exact biological replicate using ALFA-tag system is presented in Supplementary Fig. 5D–I. Source data are provided as source data file.

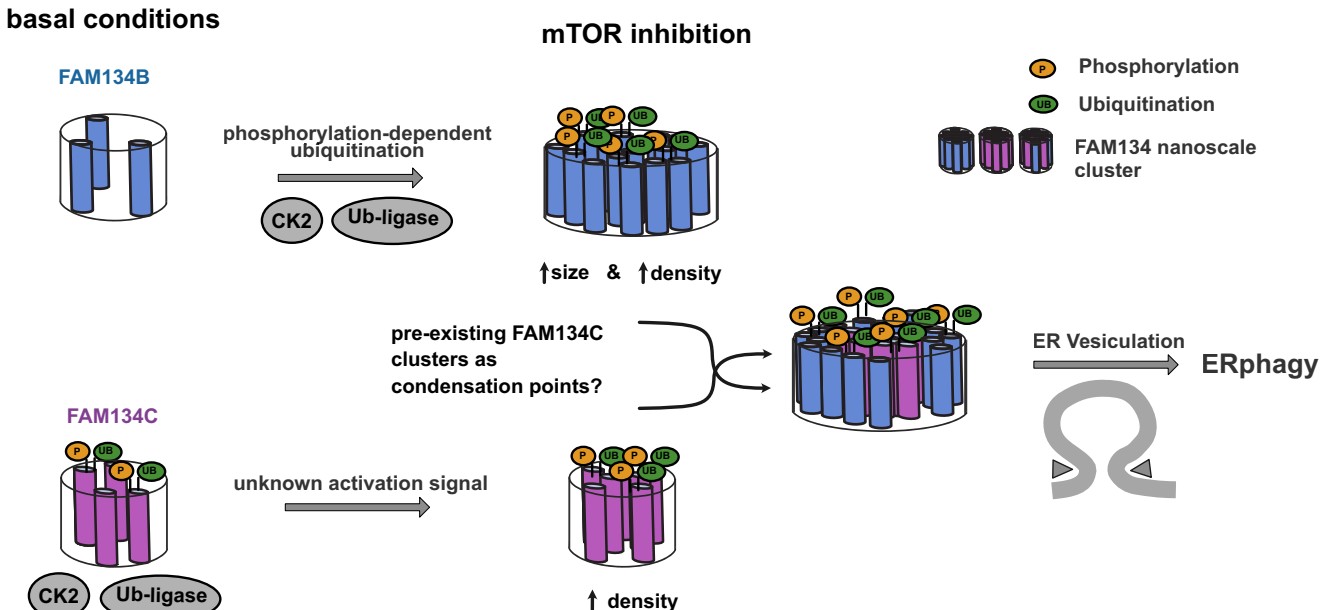

**Fig. 6 | FAM134B- and FAM134C-dependent ER-phagy induction is regulated by early interdependent post-translational modification (PTMs).** Schematic representation of the model of the study: induction of ER stress via mTOR inhibition triggers phosphorylation of FAM134B and FAM134C, which primes for their ubiquitination. Both PTMs are essential for the oligomerization process of both FAM134s and control the size and density of the clustering events leading to ER budding and ER-phagy induction. Figure created with Affinity Designer by Serif.

basal conditions. Subdividing nanoscale clusters into small (30–90 nm), medium (90–150 nm) and large clusters (>150 nm) and plotting the relative frequency distribution of localizations per cluster refined the distribution of high-density within clusters of the same size (Supplementary Fig. 5B, C). All findings could be reproduced using ALFA-tagged FAM134 proteins (Supplementary Fig. 5F–I).

Together, our data reflect higher FAM134B and FAM134C cluster densities after Torin1-induced phosphorylation and hint at a dynamic process, where FAM134 clusters reach a certain size (>90 nm) and then get further activated (by phosphorylation-dependent ubiquitination) to increase the density of included molecules. Consistent with this hypothesis, nanoscale clusters formed by phospho-mutants of FAM134B and FAM134C were able to reach a size of >150 nm, but were unable to further mature to respective high-density clusters (Fig. 5E–J). Combining this with our findings that inhibition of CK2 after Torin1 treatment suppresses ER-phagy flux (Fig. 2A, B), we predict that only high-density clusters are functional active ER-phagy sites that subsequently support transport of ER-fragments to the lysosome (Fig. 6).

## Discussion

One of the challenging questions in the field of ER-phagy is how the pathway is fine-tuned and controlled, starting from the initial trigger to bind cargo and initiate ER curvature toward ER fragmentation up to the final delivery to the lysosome. It is obvious that ER-phagy receptors play a critical role along this path, but the molecular mechanisms that drive their activation and functions remain poorly understood. In this manuscript, we provide evidence that CK2-mediated phosphorylation within the RHD domain and subsequent phosphorylation-mediated ubiquitination of FAM134B and FAM134C are essential for the formation of high-density clusters of FAM134 proteins on the ER membrane and FAM134-driven ER-phagy.

These findings originate from our previous report that FAM134C is in an inactive state under basal conditions and can serve as an enhancer of the FAM134B-dependent ER-phagy flux after stress[11]. We hypothesized that a phosphorylation cascade upon mTOR inhibition can serve as an activation signal for FAM134-driven ER-phagy. An unbiased screen against a chemical library of kinase inhibitors

modulating ER-phagy flux revealed ATR, ATM, Chk1, Akt, and CK2 to be downstream of and essential for induction of FAM134B- and FAM134C-driven ER-phagy upon mTOR inhibition. Furthermore, we identified potentially dynamically regulated phosphorylation sites after Torin1 treatment within the RHD domain and adjacent regions of FAM134B and FAM134C. In line with our chemical screening, the prediction of the target site suggests that CK2 can phosphorylate FAM134 proteins directly, which we further showed in vitro using purified proteins. CK2 is known to be ubiquitously expressed and to be a constitutively active kinase that regulates several biological processes[25]. Additionally, CK2 has been linked to autophagy-related diseases such as Alzheimer, Parkinson and Huntington disease[26]. In this study we attributed CK2 a role in orchestrating FAM134B- and FAM134C-mediated ER-phagy by phosphorylating specific sites within the RHD domain that drive the clustering of ER-phagy receptors and promote ER-phagy flux. The phosphorylation of FAM134B and subsequent induction of its oligomerization were previously reported and at that time linked to CaMKII activity[16]. Within our settings, CaMKII inhibition performed poorly in terms of preventing FAM134B-driven ER-phagy, while chemical inhibition or knockdown (KD) of CK2 had a strong effect. However, our findings do not rule out possible cooperation between CK2 and CaMKII kinases in fine-tuning these events, specifically since CAMKII was previously shown to form a complex with CK2[27]. Mechanistically, phosphorylation could alter the structure and create new binding interfaces both toward the membrane and the cytosolic side of FAM134 proteins and recruit downstream regulatory factors.

Very recently another study has highlighted the role of CK2 in ER-phagy[28]. In contrast to the above results, the study proposes CK2 as a negative regulator of ER-phagy. More precisely, CK2 was shown to phosphorylate a site close to the LIR motif of FAM134C and it was suggested that this phosphorylation could prevent its binding to LC3[28]. The presented in vivo and in silico data supported the idea, however, the impact of phosphorylation on the binding ability of FAM134C to LC3 was not fully characterized in detail in vitro. We closed this gap by performing isothermal titration calorimetry (ITC) measurements with synthetic FAM134C-LIR peptides in their phosphorylated and non-phosphorylated form (Supplementary Fig. 6A). Additionally, we

purified GFP-FAM134C-LIR from bacteria, phosphorylated it with purified CK2 and performed pulldown experiments with purified GST-LC3B (Supplementary Fig. 6B–E). Both experimental setups suggest that the FAM134C-LIR behaves as a classical phospho-LIR[29], where phosphorylation close to the LIR motif enhances binding to LC3 and therefore promotes ER-phagy. These contrasting evidences suggest a dynamic role of phosphorylation in ER-phagy and hence needs to be looked at in greater detail. We propose that the discrepancy between the in vivo and in vitro data arises from an additional, yet unknown, regulatory factor with a high affinity for phosphorylated FAM134-LIR that is able to outcompete LC3 in binding. This hypothesis is specifically intriguing as it would allow organisms to selectively regulate FAM134-driven ER-phagy in different cell types and tissues, while using the commonly expressed and constitutively active CK2 kinase as a mediator.

In addition of being phosphorylated, we show that FAM134B and FAM134C are also ubiquitinated in an mTOR- and CK2-dependent manner. Phosphorylation of FAM134 proteins may therefore create a new cytosolic binding surface important for the recruitment and/or activation of a ubiquitin ligase. The accumulation of ubiquitinated FAM134 proteins upon autophagy inhibition (BafA1 treatment) and the lack of ER-phagy capacity upon Torin1-induction in an overexpression system as well as at the endogenous level indicate that the ubiquitination of FAM134 proteins has functional relevance for FAM134-driven ER-phagy. Therefore, we hypothesized that ubiquitination is needed for the regulation and/or activation of FAM134-driven ER-phagy. In agreement with this hypothesis, inhibition of the ubiquitination system (treatment with E1 inhibitor) suppressed FAM134-driven ER-phagy flux upon mTOR inhibition. Treatment with the CK2 inhibitor abolished ubiquitination of FAM134B and FAM134C, indicating that ubiquitination is downstream of and dependent on CK2-mediated phosphorylation of FAM134.

Previous studies suggested that super-clustering of RHD-containing receptors, e.g., Atg40 in yeast and FAM134B in mammals, generates high-curved regions that subsequently can pinch-off the ER and get delivered to the lysosome[12,19]. Our data implies an interesting chain of reactions where phosphorylation and subsequent ubiquitination are involved in the regulation and formation of such super-receptor-clusters. Our data are further supported by a manuscript published during the revision of this study and reporting on the role of ubiquitination in ER-phagy regulation[30].

To dissect the consequences of PTMs on oligomerization, we analyzed two parameters of nanoscale clusters (50–200 nm) of either WT or phospho-mutants of FAM134 proteins by super-resolution microscopy: changes in cluster diameter (size) and differences in localizations within a cluster of a specific size (density). The nanoscale clusters of FAM134B WT proteins changed from unimodal (DMSO/basal) to bimodal distribution after induction of ER-phagy (Torin1). The emerging population after induction clustered around 104 nm ($\sigma = 0.24$), indicating that a cluster size of >90 nm is necessary for FAM134B-driven ER-phagy. Since the appearance of the larger population of FAM134B clusters was suppressed by CK2 inhibition as well as mutation of phosphorylation sites, phosphorylation and/or subsequent ubiquitination are essential for its formation. On the contrary, the size distribution of the clusters built by FAM134C WT or phospho-mutants was similar under all tested conditions.

We also found that the density of FAM134 molecules in clusters >90 nm started to vary dramatically, both in FAM134B and FAM134C-formed clusters in a CK2-dependent manner. This finding hints at a mechanism where clusters first grow to a certain size and then further mature by increasing the density of FAM134 molecules within a cluster. An increase in the local concentration of FAM134 proteins may be necessary to sufficiently bend the ER membrane to initiate vesiculation. Along this line, the maturation to high-density FAM134B clusters increased ~3 times after ER-phagy induction (2% vs. 5.6% relative abundance) and was greatly suppressed by treatment with CK2 inhibitor (3.7%). Interestingly, FAM134C was already forming high-density clusters under basal conditions (6.6%) and their relative abundance only slightly increased upon induction of ER-phagy (7.3%). At the same time, co-treatment with the CK2 inhibitor considerably reduced its abundance to levels below basal conditions (2.6%), indicating that FAM134C oligomerization is also regulated by CK2-dependent phosphorylation. The appearance of high-density clusters mirrors the ubiquitination status of both FAM134 proteins: the ubiquitination level of FAM134B is low under basal conditions and strongly increases upon Torin1 treatment, while FAM134C is already ubiquitinated under basal conditions and its ubiquitination level only increases slightly upon inhibition of mTOR.

Together, our data show that phosphorylation by CK2 upon mTOR inhibition, as well as phosphorylation-dependent ubiquitination of FAM134 proteins play essential roles in the regulation and activation of FAM134B- and FAM134C-driven ER-phagy. In this context, we suggest that only high-density nanoscale clusters >90 nm and >100 locations are functionally active during ER-phagy. For future studies, we hypothesize that phosphorylation-dependent ubiquitination facilitates the formation of high-density FAM134 clusters, presumably by introducing conformational changes between adjacent FAM134 molecules and/or by recruiting cytosolic factors with oligomerizing properties to the forming ER vesicle. As FAM134C serves as an enhancer of FAM134B[11], we further hypothesize that pre-existing, ubiquitinated FAM134C nanoscale clusters may serve as collection points for ER-phagy substrates and subsequently as nucleation points for FAM134B and that this is the reason behind the rapid occurrence of a second, larger FAM134B cluster population upon Torin1 treatment. It is also interesting to note that in our system overexpression of FAM134C has a greater impact on ER-phagy flux than FAM134B, which may imply that FAM134C is also an enhancer for other, endogenously expressed ER-phagy receptors.

## Methods

### Reagents and antibodies

Kinase inhibitor library was provided by the SGC Frankfurt donated probes program and complemented by additional kinase inhibitors. Torin1 was purchased from Bio-techne, 4247 and BafA1 from Cayman Chemical Corporation, 11038. Following primary antibodies were used for immunoblotting and immunofluorescent staining: rabbit anti-HA tag (Cell signaling Technology, 3724T) used at 1/2000 dilution for immunoblotting, rat anti-HA tag (Roche, 11867432001) used at 1/5000 dilution for immunofluorescence, rabbit anti-FAM134B (Sigma Prestige, HPA012077) used at 1/1000 dilution for immunoblotting, rabbit anti-FAM134C (Sigma Prestige, HPA016492) used at 1/1000 dilution for immunoblotting, mouse anti-REEP5 (Santa Cruz Biotechnology, sc-393508) used at 1/1000 dilution for immunofluorescence, rabbit anti-GST (Cell Signaling Technology, 2625S) used at 1/2000 dilution for immunoblotting, mouse anti-GFP (Santa Cruz Biotechnology, sc-9996) used at 1/2000 dilution for immunoblotting, rabbit anti-CK2 (Cell signaling Technology, 2656S) used at 1/1000 dilution for immunoblotting, mouse anti-ubiquitin P4D1 (abcam, ab7254) used at 1/1000 dilution for immunoblotting, rabbit anti-GAPDH (Cell Signaling Technology, 2118L) used at 1/1000 dilution for immunoblotting. Following secondary antibodies were used at 1/10000 dilution for immunoblotting: anti-mouse (Thermo Fisher Scientific, 31326) anti-rabbit (Thermo Fisher Scientific, 32460) anti-rat (Cell Signaling Technology, 70775). Following secondary antibodies were used at 1/500 dilution for immunofluorescent staining and super-resolution microscopy: anti-mouse Alexa Fluor® 532 (Life technologies, A11002), anti-rabbit (AffiniPure, 711-005-152) and anti-mouse (AffiniPure, 115-005-003), covalently labeled with short oligonucleotide strands anti-P1 (5′-ATCTACA-TATT-3′) and anti-P5 (5′-TATGTAACTTT-3′), respectively.

## Plasmids

pCW57-CMV-ssRFP-GFP-KDEL was a gift from Noboru Mizushima (University of Tokyo, Tokyo, Japan), (Addgene plasmid # 128257; RRID: Addgene_128257). pLenti CMV GFP Hygro (656-4) was a gift from Eric Campeau and Paul Kaufman (Addgene plasmid # 17446; http://n2t.net/addgene:17446; RRID: Addgene_17446). pLenti CMV GFP Hygro (656-4) was used as the backbone to create pLenti-CMV-Hygro- ssRFP-GFP-KDEL for consecutively expressing ssRFP-GFP-KDEL. Briefly, pLenti CMV GFP Hygro (656-4) was digested with two restriction enzymes (BamHI and SalI) to remove the GFP cassette. ssRFP-GFP-KDEL was then amplified by PCR and cloned into pLenti CMV GFP Hygro (656-4) plasmids using Sequence and Ligation Independent Cloning (SLIC) with T4 DNA polymerase. iTAP-FLAG-HA-FAM134B (also referred to as pMCSV-puro-FLAG-HA-FAM134B) and iTAP-FLAG-HA-FAM134B (also referred to as pMCSV-puro-FLAG-HA-FAM134C) were gifts from Ivan Dikic (Goethe University Frankfurt am Main, Germany). iTAP-FLAG-HA-FAM134B and iTAP-FLAG-HA-FAM134C were used as the backbone to create different phosphorylation mutants, as well as the ALFA-tag expressing cells using quick-change PCR mutagenesis. The FLAG sequence from the iTAP-FLAG-HA-FAM134B and iTAP-FLAG-HA-FAM134C vectors was replaced by the sequence of ALFA-tag to create vectors expressing ALFA-tagged- FAM134B and FAM134C. The amino acid sequence of the ALFA-tag (SRLEEELRRRLTE) was used based on the previous publication[23].

## Cell culture

U2OS T-REx and HEK293T cells were maintained in a 37 °C incubator with humidified 5% CO$_2$ atmosphere and cultured in Dulbecco's modified Eagle's medium (DMEM) (Thermo Fisher Scientific) supplemented with 10% fetal bovine serum (FBS) (Thermo Fisher Scientific) and 100 U/ml (1%) penicillin/streptomycin (PS) (Thermo Fisher Scientific). U2OS T-REx cells stably expressing ssRFP-GFP-KDEL and inducibly individual FAM134 proteins were described previously[11]. These cells were treated with 1 µg/ml DOX (Sigma-Aldrich, D9891) to induce FAM134 protein expression. Prior to each treatment, cells were plated the day before in order to perform the experiments when cells had a final confluency of 70–80%. HEK293T was purchased from ATCC and was used for transient transfections. For transient expression, DNA plasmids were transfected with JetPEI (Polyplus, 101000020) according to manufacturer instructions.

## Cell line generation

U2OS T-REx cells were a gift from Stephen Blacklow (Brigham and Women's Hospital and Harvard Medical School)[31] and were culture using DMEM supplemented with 10% FBS, 1% PS and 15 µg/ml blasticidin S (Thermo Fisher scientific, A1113902). Lentiviral particles were produced in HEK293T cells using pLenti-CMV-Hygro-ssRFP-GFP-KDEL and pMD2.G and psPAX2. Retroviral particles were produced in HEK293T using iTAP-FLAG-HA or iTAP-ALFA-HA plasmids expressing FAM134B, FAM134C or different phosphorylation mutants, packaging plasmid (pCMV-Gag-Pol) and the envelope plasmid (pCMV-VSV-G). The U2OS T-REx cells were plated and propagated in 6-well plates overnight. On the next day, transduction with viruses was performed. Two days later, transduced cells with iTAP-FLAG-HA or iTAP-ALFA-HA plasmids were selected using 15 µg/ml blasticidin S and 2 µg/ml puromycin (Invitrogen, Thermo Fisher Scientific, Waltham, Massachusetts, USA) for 3 passages. U2OS T-REx cells expressing FAM134B, FAM134C or different mutants were used for transduction with ssRFP-GFP-KDEL retroviruses. The transduced cells were selected using 15 µg/ml blasticidin S, 2 µg/ml puromycin and 150 µg/ml hygromycin B. Afterward, the cells were sorted for different populations of RFP- and GFP-expressing cells using SH800S Cell Sorter (Sony Biotechnology, San Jose, CA, USA). Stable clones were maintained in the same medium as used for selection.

*Fam134b* and *Fam134c* knockout MEFs were provided by Prof. Christian Huebner (Jena University) and were culture using DMEM

supplemented with 10% FBS. These cells were obtained from *Fam134b* or *Fam134c* single knockout mice and have been reconstituted with cDNAs cloned in a retroviral pBABE plasmid. Retrovirus were generated in HEK293T cells co-transfecting the retroviral vector, containing the selected cDNA, with the packaging plasmid (viral Gag-Pol) and the envelope plasmid (VSV-G). Lentivirus were also generated co-transfecting HEK293T cells with the pCW57-CMV-ssRFP-GFP-KDEL (Addgene #128257) vector together withpPAX2 packaging plasmid and pMD2.G envelope plasmid.

## Protein purification

The recombinant GFP-FAM134C-LIR protein was overexpressed in *E. coli* strain BL21(DE3). Briefly, primary cultures were grown overnight followed by secondary cultures in LB media containing appropriate antibiotics at 37 °C. The secondary cultures of *E.coli* were grown up to 0.6 OD$_{(600)}$ and induced the cells with 0.25 mM IPTG for 18 h at 18 °C. The cultures were harvested by centrifugation at 20,000 × *g* for 8 min. Resulting cell pellets were lysed using sonicator. The 6X-Histidine tagged proteins were purified using the Ni-NTA affinity chromatography method using solution containing 200 mM NaCl and 100 mM Tris pH 8.0. An imidazole gradient of 20 mM to 250 mM was used to elute the bound protein. Eluted fractions containing protein were pooled and concentrated to reach an appropriate concentration and stored at −20 °C in buffer containing 50% glycerol. Purification of the FAM134B-RHD domain has been published before[30].

## Endogenous pulldown of FAM134 proteins

HEK293T cells were plated in 15 cm dishes and subjected to following treatments (DMSO (Sigma-Aldrich), and 250 nM Torin1 (Bio-techne) in combination with 1 µM CK2 inhibitor (SGC donated probes). Three copies of each individual condition were harvested and pulled to be then lysed in 1 ml of ice-cold RIPA buffer: 10 mM Tris-HCl, (pH 8.0), 1 mM EDTA, 0.5 mM EGTA, 1% (v/v) Triton X-100, 0.1% SDS, 140 mM NaCl, cOmplete protease inhibitor cocktail (Roche), phosphatase inhibitor cocktail 1 and 2 (Sigma-Aldrich). Lysates were cleared by centrifugation at 20,000 × *g* for 15 min at 4 °C. Three mg of proteins were subsequently incubated for 1 h with 10 µl of FAM134B or FAM134C antibodies (Sigma Prestige-HPA012077 and -HPA016492, respectively) and then overnight at 4 °C with A/G PLUS agarose beads (Santa Cruz Biotechnology) with constant mixing. Beads were then washed three times in wash buffer (10 mM Tris-HCl (pH 8.0), 1 mM EDTA, 0.5 mM EGTA, 1% (v/v) Triton X-100, 0.1% SDS, 250 mM NaCl, cOmplete protease inhibitor cocktail (Roche), phosphatase inhibitor cocktail 1 and 2 (Sigma-Aldrich)), resuspended in SDS sample buffer and boiled at 95 °C for 10 min. Beads were collected by centrifugation at 2500 × *g* for 2 min and supernatants were loaded on SDS-PAGE and western blot was performed using the indicated antibodies.

## In vitro and in cellulo GFP-Trap pulldown

For in cellulo experiment, HEK293T cells were transiently co-transfected with two constructs carrying either V1-FAM134B and V2-FAM134B or V1-FAM134C and V2-FAM134C each independently tagged with half GFP at the N-terminus. Following 6 h treatment with the indicated conditions, cells were washed with 1x PBS and lysed in ice-cold RIPA buffer: 10 mM Tris-HCl, (pH 8.0), 1 mM EDTA, 0.5 mM EGTA, 1% (v/v) Triton X-100, 0.1% SDS, 140 mM NaCl, cOmplete protease inhibitor cocktail (Roche), phosphatase inhibitor cocktail 1 and 2 (Sigma-Aldrich) and deubiquitinase inhibitor (N-ethylmaleimide, NEM). Samples were incubated on ice for 15 min and centrifuged at 20,000 × *g* for 15 min at 4 °C. GFP-Trap beads (Chromotek) were equilibrated and washed three times in 500 µl of ice-cold RIPA buffer and were spun down at 2500 × *g* for 2 min at 4 °C. In total, 1 mg of lysates was added to 5 µl of equilibrated GFP-Trap beads and incubated overnight 4 °C with constant mixing. Tubes were spun at 2500 × *g* for 3 min at 4 °C. GFP-Trap beads were washed three times with 500 µl ice-

cold lysis buffer. Fifty μl of 1X SDS sample buffer was then added to the GFP-Trap beads and were boiled for 10 min at 95 °C. Beads were collected by centrifugation at 2500 × *g* for 2 min and SDS-PAGE was performed with the supernatant.

For in vitro experiment, purified proteins (1 μM GST-LC3B, 1 μM GFP-FAM134C-LIR or 1 μM of phosphorylated GFP-FAM134C-LIR) were mixed with 100 mM NaCl, 100 mM Tris pH 7.5 and 5 μl of equilibrated GFP-Trap beads (Chromotek) and incubated overnight at 4 °C with constant mixing. Samples were then washed three times with 500 μl ice-cold wash buffer (300 mM NaCl, 100 mM Tris pH 7.5) and processed as described in the in cellulo experiment.

### Tandem ubiquitin binding entities 2 (TUBE 2) assay

U2OS cells expression FAM134 proteins priorly subjected or not to KD experiments were induced with 24 h DOX (1 μg/ml) treatment. Following treatments (DMSO, Torin1, CK2 inhibitor, BafA1), confluent 10 cm dishes of U2OS cells were harvested in 500 μl of ice-cold lysis buffer: 50 mM Tris-HCl (pH 8.0), 120 mM NaCl, 1% (v/v) NP-40, cOmplete protease inhibitor cocktail, phosphatase inhibitor cocktail 1 and 2. Lysates were cleared by centrifugation at 20,000 × *g* for 15 min at 4 °C and 1 mg of proteins was incubated overnight at 4 °C with 10 μl of agarose-TUBE 2 resin. Reactions were rotated overnight at 4 °C, washed three times with NP-40 buffer and analyzed by SDS-PAGE. Anti-HA antibody was used to detect HA-tagged FAM134 proteins and ubiquitinated species.

### Immunoblotting of total cell lysates

Following KD experiment, U2OS cells were harvested after removing the culture medium and washing with 1X PBS and lysed in lysis buffer (150 mM NaCl, 50 mM Tris-HCl pH 7.5, 10 mM NaF, 1 mM EGTA, 5 mM MgCl₂, 0.1% SDS) supplemented with protease and phosphatase inhibitors. Insoluble cell components were separated at 20,000 × *g* for 15 min at 4 °C and total protein concentration was assessed using BCA assay (Thermo Fisher Scientific). Protein denaturation was performed in SDS sample buffer at 95 °C for 10 min.

Protein lysates were resolved in SDS-PAGE gels and transferred to PVDF (Millipore, 0.2 μm) membrane. Membranes were saturated in PBST 0.1% Tween-20 containing 5% BSA and incubated overnight at 4 °C with the specific primary antibody.

### In cellulo mass spectrometry identification of phosphoserines

U2OS cell lines stably expressing FAM134 proteins were cultured in 15 cm dishes up to 70% confluency in the presence of 1 μg/ml DOX prior to any treatment. Each replicate from every condition consisted of 2 × 15 cm dishes lysates (50 mM Tris PH 7.4, 150 mM NaCl, 1% NP-40, fresh protease and phosphatase inhibitors) pooled together. The purification of FAM134 proteins from each condition was performed via anti-HA antibody-conjugated beads. To remove as many impurities and weak interactors as possible, 6 washing steps were performed. For the first two washes, 50 mM Tris PH 7.4, 500 mM NaCl, 1% NP-40 containing phosphatase inhibitor and Roche protease inhibitor tablets was used. The following two washes were performed with 50 mM Tris PH 7.4, 150 mM NaCl and all inhibitors. The last two washes were performed with 1X PBS to remove all traces of detergent prior to MS processing. Beads were incubated with SDC buffer (2% SDC, 1 mM TCEP, 4 mM CAA, 50 mM Tris pH 8.5) at 65 °C for 30 min. Proteins were digested with 500 ng of trypsin (Promega) and LysC (Wako Chemicals). Digestion was stopped with 1% trifluoroacetic acid (TFA) in isopropanol. Following digestion, SDB-RPS columns with a capacity of 10 μg/14 G plug were utilized for sample clean up. Samples were analyzed on a Q Exactive HF coupled to an easy nLC 1200 (Thermo Fisher Scientific) using a 35 cm long, 75 μm ID fused-silica column packed in house with 1.9 μm C18 particles (Reprosil pur, Dr. Maisch), and kept at 50 °C using an integrated column oven (Sonation). Peptides were eluted by a non-linear gradient from 4-28% acetonitrile over 45 min and directly sprayed into the mass-spectrometer equipped with a nanoFlex ion source (Thermo Fisher Scientific). Full scan MS spectra (350–1650 *m/z*) were acquired in Profile mode at a resolution of 60,000 at *m/z* 200, a maximum injection time of 20 ms and an AGC target value of 3 × 10⁶ charges. Up to 10 most intense peptides per full scan were isolated using a 1.4 Th window and fragmented using higher energy collisional dissociation (normalized collision energy of 27). MS/MS spectra were acquired in centroid mode with a resolution of 30,000, a maximum injection time of 110 ms and an AGC target value of 1 × 10⁵. Single charged ions, ions with a charge state above 5 and ions with unassigned charge states were not considered for fragmentation and dynamic exclusion was set to 20 s.

### In vitro mass spectrometry identification of phosphoserines

Three technical replicates containing each 1 μM of the purified RHD region of FAM134B with 1 μM of each purified subunit of CK2 (CSNK2A1 and CSNK2A2) were incubated for 2 h at 37 °C. ATP was added at a final concentration of 10 mM and mixed with CK2 buffer (20 mM Tris-HCl pH 7.5, 50 nM KCl, 10 mM MgCl₂). The negative controls had no CK2 added to them. Samples were processed and analyzed for MS as previously discussed with the exception of a lower amount of trypsine/LysC (200 ng).

### Mass spectrometry data processing and analysis

MS raw data processing was performed with MaxQuant (v 2.1.0.0) and its in-build label-free quantification algorithm MaxLFQ applying default parameters[32]. Acquired spectra were searched against the human reference proteome (Taxonomy ID 9606) downloaded from UniProt (21-05-2022; 82.492 protein count), self-made fasta file containing the precise sequence of the RHD construct used for the in vitro experiment, and a collection of common contaminants (244 entries) using the Andromeda search engine integrated in MaxQuant[33]. Identifications were filtered to obtain false discovery rates (FDR) below 5% for both peptide spectrum matches (PSM; minimum length of 7 amino acids) and proteins using a target-decoy strategy[34].

The MS/MS spectra for the modified and best-localized peptides were taken from the visualization option of MaxQuant software. This information is included in source data file as "Source_data_MS_Spectra". All the tables in Supplementary Fig. 3 show the different confidence scores, posterior error probability (PEP), and localization probability. It also shows the positions they hold within the WT full-length FAM134B protein and in which samples the peptides were found.

### ER-phagy flux assay

U2OS T-REx FLAG-HA-FAM134s stable and inducible cell lines were infected with lentivirus carrying the pCW57-CMV-ssRFP-GFP-KDEL (Addgene #128257). We previously deleted the tetracycline response element, via PCR, from ssRFP-GFP-KDEL in order to have constitutive expression. Wild-type and *Fam134*s KO MEFs were infected with pCW57-CMV-ssRFP-GFP-KDEL (Addgene #128257) in order to generate ER-phagy reporter inducible cell lines. Fluorescence of GFP and RFP, as well as cell confluence (phase), were monitored over time via the IncuCyte® S3 (Sartorius, Germany) in 384-well format. In total, 1500 cells (U2OS) per well were seeded in 50 μl DMEM media, supplemented with 10% FBS, 1% PS, 2 μg/ml puromycin, 15 μg/ml blasticidin S, 1 μg/ml DOX and incubated for 24 h before treatment. For indicated treatments, 50 μl of media containing indicated treatments were added. Between the actual treatment and the first scan point (0 h) ~5–10 min passed for technical reasons. Screens in all three channels were taken at indicated time points following the treatment. ER-phagy flux was monitored through changes in the ratio of the total fluorescence intensity of RFP/GFP. Each point represents the averaged ratio of data obtained from three individual wells (technical replicates).

## Antibody-oligonucleotide conjugation for DNA-PAINT

Secondary antibodies donkey anti-rabbit (AffiniPure, 711-005-152) and goat anti-mouse (AffiniPure, 115-005-003) were covalently labeled with short oligonucleotide strands anti-P1 (5′-ATCTACATATT-3′) and anti-P5 (5′-TATGTAACTTT-3′), respectively, via DBCO-sulfo-NHS ester chemistry. In brief, concentrated secondary antibodies were incubated with a 20-fold molar excess of DBCO-sulfo-NHS ester (Jena Bioscience, CLK-A124-10) for 90 min at 4 °C. Excess reagent was removed using spin filters (Merck, Germany) and docking strand labeled antibodies were incubated with a tenfold molar excess of azide-functionalized DNA-docking strands overnight at 4 °C. Samples were then filtered through Amicon centrifugal filters (100,000 MWCO) to remove unbound oligonucleotides and change storage buffer to PBS. Finally, antibody-DNA complexes were concentrated to 5 mg/ml and stored at 4 °C.

## Sample preparation for DNA-PAINT

For DNA-PAINT experiments, U2OS cells were seeded in fibronectin-coated Ibidi μ-slide VI chambers at a density/confluence of 60% and FAM134s protein expression was induced with 1 μg/ml DOX overnight. Cells were chemically fixed with prewarmed (37 °C) 4% methanol-free formaldehyde (Sigma-Aldrich, Germany) in PBS for 20 min followed by three washing steps with PBS. Primary antibodies rabbit anti-HA (1:500) was subsequently added in permeabilization/blocking buffer (10% fetal calf serum (FCS) (Gibco), 0.1% saponin (Sigma-Aldrich)) and incubated for 60 min at room temperature. Excess primary antibodies were removed from the chambers by three washing steps with PBS and custom docking-strand labeled secondary antibodies (1:100 in permeabilization /blocking buffer) were added to the chambers. After incubation for 60 min at room temperature, samples were washed thrice to remove excess secondary antibodies. Finally, samples were post-fixed using 4% methanol-free formaldehyde in PBS for 10 min at room temperature followed by three washing steps with PBS. For colocalization experiments, fiducial markers were used for image registration (125 nm gold beads (Nanopartz, USA)). Gold beads were sonicated for 10 min and diluted 1:30 in PBS. Diluted samples were again sonicated for 10 min and 100 μl of the solutions were added to the microscopy chambers. After settlement of the gold beads (5 min), the samples were washed thrice with PBS. Cell chambers were finally assembled to a microfluidic device (Bruker, USA) for automated buffer exchange. Prior to image acquisition, ATTO-655 labeled imager Strands P1 or P5 (400 pM, 0.5 M NaCl, PBS, pH 8.3) were injected to the flow chamber at a flow rate of 600 μl/min. For colocalization experiments, imager strands were exchanged by washing the samples with 1 ml PBS followed by injection of new imager strands.

## Microscopy setup and data acquisition

DNA-PAINT based super-resolution microscopy was performed at the N-STORM super-resolution microscopy system (Nikon, Japan), equipped with an oil immersion objective (Apo, 100x, NA 1.49) and an EMCCD camera (DU-897U-CS0-#BV, Andor Technology, Ireland) as described previously[35]. Fluorophore-conjugated oligonucleotides were excited with a collimated 647 nm laser beam at an intensity of 2 kW/cm$^2$ (measured at the objective) at highly inclined and laminated optical sheet (HILO) mode. In total, 40,000 and 20,000 consecutive frames were acquired for imager strands P1 (5′-TAGATGTAT-3′-ATTO655) and P5 (5′-CATACATTGA-3′-ATTO655), respectively. Image acquisition was performed at 20 Hz (P1) and 10 Hz (P5) in active frame transfer mode with an EMCCD gain of 200, a pre amp gain of 1 and at an effective pixel size of 158 nm. Software tools NIS Elements (Nikon, Japan), LCControl (Agilent, USA), and Micro-Manager 1.4.22[36] were used for control of the optical setup and image acquisition. FAM134B and FAM134C were imaged sequentially using a microfluidic assisted exchange of imager strands. The spatial resolution of our super-resolution images was determined to 22 ± 2 nm (mean ± s.d.[37], allowing the detection of FAM134 nanoscale cluster.

## Image processing

Localization of single molecules and reconstruction of super-resolution images were performed with modular software package Picasso[38]. Integrated Gaussian maximum likelihood estimation was used for detection of single molecules using the following parameter: Min. net. gradient of 15,000, a baseline of 205, a sensitivity of 4.78, a quantum efficiency of 0.95. Sample drift correction and image alignment were performed via redundant cross-correlation algorithm using fiducial marker. For FAM134B (WT and S149,151,153A) and FAM134C (WT and S258A), localizations were subsequently filtered based on ATTO-655 single-molecule footprint (PSF symmetry 0.6 < FWHM(x)/FWHM(y) < 1.4). Signals from the same origin detected in consecutive images were linked within a radius of five times the nearest-neighbor based analysis (NeNa)[39] localization precision and a maximum dark time of eight consecutive frames. Remaining signals from fiducial marker were removed by excluding traces from the same origin with a length larger than 20 consecutive frames and a min. net gradient <100,000. To account for the high expression level in FAM134C WT expressing cells, nanoscale cluster were first segmented from ER-located background signal via Density-based spatial clustering and application with noise algorithm[40] analysis using a radius of 16 nm and a minimum density of 30 localizations followed by data filtering as described previously.

## Cluster analysis

DBSCAN algorithm was used to identify protein nanocluster in DNA-PAINT super-resolution images. A radius of 32 nm (FAM134B WT and S149,151,153A and FAM134C S258A) and a minimum density of 10 localizations were used as parameter. For FAM134C WT, a radius of 16 nm and a minimum density of 5 localizations were used for DBSCAN analysis. For the determination of nanoscale cluster density in U2OS cells, protein clusters were normalized with respect to the cell area. Therefore, cell contours were drawn in super-resolution images using ImageJ and the cell area was determined.

## Sample preparation for STED imaging

Sample preparation for stimulated emission depletion (STED) microscopy was as described for exchange DNA-PAINT in material and methods with the exception of the antibody used. ALFA-FAM134s were stained directly using a nanobody against ALFA tag coupled to a Abberior Star 635P fluorophore (FluoTag®-X2 anti-ALFA−Abberior-Star635P, NanoTag Biotechnologies GmbH) diluted 1:500 in permeabilization/blocking buffer (10% FCS (Gibco), 0.1% saponin (Sigma-Aldrich)) and incubated for 60 min at room temperature.

## Sample preparation for DNA-PAINT

Sample preparation for DNA-PAINT was the same as described for exchange DNA-PAINT in material and methods with the exception of the antibody used. ALFA-FAM134s were targeted directly using an anti-ALFA nanobody carrying a 5xR1 docking strand (5-TCCT CCTCCTCCTCCTCCT-3′) (Massive-TAG-Q Anti-ALFA Custom, Massive Photonics, GmbH) which was diluted in permeabilization/blocking buffer (10% FCS (Gibco), 0.1% saponin (Sigma-Aldrich)) at a dilution of 1:500 and incubated for 60 min at room temperature.

## Microscopy setup and data acquisition

DNA-PAINT based super-resolution microscopy was performed using an N-STORM super-resolution microscopy system as described in material and methods. In total, 25,000 consecutives frames were acquired for the imager strand R1 (5′-AGGAGGA-3′-ATTO655) at 10 Hz using settings as described in "Material and methods".

## Image processing

Localization of single molecules and reconstruction of super-resolution images were performed with modular software package

Picasso as described in material and methods with no additional filtering for FAM134C.

## Cluster analysis

DBSCAN algorithm was used to identify protein nanocluster in DNA-PAINT super-resolution images using a radius of 32 nm and a minimum density of 5 localizations (ALFA-FAM134B) or 10 localizations (ALFA-FAM134C). Only clusters that contain more than 20 localizations were used for further analysis. All cluster with a mean frame (mean number of frames in which localizations occurred) in the first or last 30% of the total frame number and a standard deviation $20\% < \sigma < 35\%$ were excluded from the analysis.

## Isothermal titration calorimetry

For ITC experiments, LC3B was obtained by Ub-fusion technology based on protocols described previously[41]. Synthetic peptides were purchased from the Tufts University Core Facility (0.1 mmol scale, HPLC-purified). Before experiments, proteins and peptides were equilibrated in buffer containing 25 mM Hepes pH 7.5, 100 mM NaCl. All titration experiments were performed in triplicates at 25 °C using a VP-ITC microcalorimeter (Malvern Instruments Ltd., UK). ITC data were analyzed using NITPIC for baseline correction and SEDPHAT for fitting. The peptides at a concentration of 250 μM were titrated into 20 μM LC3B in 25 steps. Dilution heat from titrating peptide into buffer or buffer into protein was subtracted. Concentrations of proteins and peptides were calculated from the UV absorption at 280 nm using a Nanodrop spectrophotometer (Thermo Fisher Scientific, DE; USA).

## Statistical analysis

FAM134B and FAM134C nanocluster diameter were tested for normal distribution via Shapiro−Wilk normality test and statistical significance via non-parametric Mann−Whitney $U$ test. The relative frequency distribution of nanocluster diameter was fitted with a Log-normal distribution and the mode of distribution was calculated as $e^{(\mu-\sigma^2)}$. All statistical analysis were performed in OriginPro (Origin Lab) or GraphPad Prism for quantified blots.

## Reporting summary

Further information on research design is available in the Nature Portfolio Reporting Summary linked to this article.

# Data availability

The mass spectrometry proteomics data have been deposited to the proteomeXchange Consortium via the PRIDE[42–44] partner repository with the dataset identifier PXD043003. Microscopic images not necessary for data interpretation are available under restricted reason due to the size and format of the data sets, access can be requested via contacting the corresponding author. Processed data of microscopic images are provided in this paper. Source data are provided with this paper.

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

## Acknowledgements

We thank all members of the Quantitative Proteomics Unit at IBC2 (Goethe University, Frankfurt), in particular, Rajeshwari Ratore and Marina Hoffmann for support and expertise in Proteomics methodology and data analysis, Martin Adrian-Allgood for technical support, Kristina Wagner for producing LC columns, and David Krause for help in (bio) informatics. The authors thank Stefan Knapp and Andreas Krämer for providing tools and reagents including purified CK2 and Ivan Dikic for critical comments on the text. We thank Julia Pomirska for support in the purification of the RHD domain and Sergio Alejandro Poveda Cuevas for help with the RHD domain scheme. We thank To Dinh Nguyen Ngu and Ivana Peluso for technical support. This work was funded by the Deutsche Forschungsgemeinschaft (DFG, German Research Foundation)—grant numbers 259130777 (SFB1177) to V.D., M.H. and A.S.; 515275293 (FuGG; cell sorter); 403765277 (FuGG; mass spectrometer); and 512574446 (FuGG; spinning disc confocal) to A.S., the Dr. Rolf M. Schwiete Stiftung (13/2017), the EU/EFPIA/OICR/McGill/KTH/Diamond Innovative Medicines Initiative 2 Joint Undertaking (EUbOPEN grant no. 875510) to A.S., the Telethon Foundation (grant TMPGMFU22TT) to P.G., the AIRC (grant MFAG 2020 ID 24856) to P.G., and a postdoctoral fellowship from EMBO (ALTF 199-2021) to S.K.K. Images were created in part using BioRender.

## Author contributions

R.B.: conceptualization, investigation, formal analysis, visualization, writing—original draft preparation; H.H.X.: investigation, validation, writing—review and editing; M.G.: methodology, investigation, formal analysis, visualization, writing—original draft preparation; P.S.M.: investigation, formal analysis, data curation, writing—review and editing; L.B.: investigation; T.G.: investigation, validation; S.K.K.: investigation; K.H.: investigation, formal analysis; S.C.F.: investigation, validation; V.B.: investigation; P.C.P.: investigation; A.B.: methodology, investigation; K.H.: methodology; G.T.: methodology, formal analysis; T.J.: investigation, formal analysis; M.M.: resources; A.G.: resources; V.D.: methodology, supervision; P.G.: validation, resources, supervision, writing—review and editing; M.H.: methodology, supervision, writing—review and editing, funding acquisition; A.S.: conceptualization, methodology, formal analysis, validation, data curation, visualization, writing—original draft preparation, supervision, project administration, funding acquisition.

## Funding

## Competing interests

The authors declare no competing interests.

## Additional information

[1]Institute of Biochemistry II (IBC2), Faculty of Medicine, Goethe University, Frankfurt am Main, Germany. [2]Buchmann Institute for Molecular Life Sciences (BMLS), Goethe University, Frankfurt am Main, Germany. [3]Institute of Physical and Theoretical Chemistry, Goethe University, Frankfurt am Main, Germany. [4]Institute of Biophysical Chemistry and Center for Biomolecular Magnetic Resonance, Goethe University, Frankfurt am Main, Germany. [5]Telethon Institute of Genetics and Medicine (TIGEM), Pozzuoli, Italy. [6]Institute of Molecular Biology, Mainz, Germany. [7]Department of Clinical Medicine and Surgery, Federico II University, Naples, Italy. [8]These authors contributed equally: Hung Ho-Xuan, Marius Glogger, Pablo Sanz-Martinez. ✉e-mail: stolz@em.uni-frankfurt.de

