## [Peer Review File · Nature Communications]

The function of ER-phagy receptors is regulated through phosphorylation-dependent ubiquitination pathwaysReviewers' Comments:

Reviewer #1:

Remarks to the Author:

Pathways for regulation of induction of ER-phagy via cargo receptors are not well understood. Berkane et al propose that upon nutrient stress/mTOR inhibition that CK2 activity is in fact required to promote FAM134B phosphorylation, ubiquitination and clustering. FAM134C may be regulated more basally by this mechanism, although CK2 is clearly shown to be required for FAM134C induced ER-phagy. The pharmacologic screen to identify CK2 is very nice, but the authors' conclusions are somewhat preliminary/correlative and further data is in places required to confirm their proposal.

1) The authors should also cite and address the recent paper from Settembre lab (which has at least one author in common with this submission) that, superficially at least, seems to suggest the opposite conclusion, that CK2 in fact suppresses ER-phagy via FAM134C.

2) The CK2 data is supported entirely by use of pharmacologic inhibitors, which can be off-target. Does knockout of any of the key CK2s prevent FAM134B ubiquitylation/phosphorylation and/or clustering?

3) The paper relies on overexpression of FAM134 paralogues throughout. It would be good to show that endogenous FAM134B/C clusters dependent upon CK2 activity, for example by imaging, and to show that endogenous FAM134B is ubiquitylated.

4) Phosphorylation sites are mapped in response to Torin via mass spectrometry but what is missing is direct demonstration that these are sensitive to CK2 inhibition in this analysis (I do acknowledge that some at least are a good fit for CK2 consensus sites). This is particularly important for the sites used as mutants later on to demonstrate the role of CK2-mediated phosphorylation in clustering. Does recombinant CK2 phosphorylated purified FAM134B at these sites?

5) It's unclear what Figure 2D is supposed to be showing. It feels it is overlooked in the text.

6) Fig. 3F - the majority of FAM134B/C IPd by anti-GFP, which I presume to indicate dimeric FAM134B/C, does not change upon Torin or CK2 inhibition. A minority Ub species does disappear with CK2 inhibition. So I understand that this indicates that a small amount of multimeric FAM134B/C is ubiquitylated dependent upon CK2. However, this assay requires further controls. For instance, does the split-Venus interaction occur in cellulo or post-lysis? Without knowing this, it is difficult to say the ubiquitination occurs upon multimeric FAM134

7) Figure 4/Supp Fig. 4 purport to show statistical analyses determining which cluster size distributions are probably different from one another, but I could not see the results of these analyses in the figure or the legend (e.g. p-val for Mann-Whitney U-test).

MINOR POINTS

1) Preamble paragraph at start of Results chapter seems out of place

2) Missing (REF) on page 17

3) Figure 1 - KDEL probe should be depicted within the ER lumen

4) Abstract, "serin" typo, should be "serine"

5) page 8 - ATM/ATR/Chk1 are also all involved in sensing DNA damage, this should be acknowledged

Reviewer #2:

Remarks to the Author:

In this manuscript, Berkane and co-workers demonstrate that the CK2 kinase phosphorylates the ER proteins FAM134B and FAM134C upon inhibition of mTOR (by Torin) and that this is required for their subsequent ubiquitination, clustering and function in ERphagy. CK2 was identified in a screen for drugs affecting FAM134B/C mediated ERphagy upon mTOR inhibition.

The manuscript is generally well written and the data are nicely presented. However, the data are largely descriptive and there is no link to any physiological relevance of their findings. To strengthen the manuscript the authors should address the following comments.

Specific comments:

The system used for the drug screen is based on cell lines with overexpression of both an ERphagy reporter (KDEL reporter) and HA-FLAG-FAM134B or HA-FLAG-FAM134C (FAM134C) that are treated with Torin to induce ERphagy. As overexpression of FAM134B/C in itself will induce ERphagy, this system might be somewhat artificial. Although the authors do show that the CK2 inhibitor also has an effect on Torin-mediated ERphagy in control cells, it will be important to show directly that this is due to CK2-mediated phosphorylation of FAM134B/C. They need to demonstrate that CK2 does phosphorylate FAM134 and that expression of corresponding phospho-mutants are defect in ERphagy. They should also demonstrate that knock-down of CK2 inhibits ERphagy to prove specificity of the inhibitor.

The authors claim that CK2 mediated regulation of FAM134s and ERphagy is regulated downstream of mTOR. They should use other mTOR modulators (e.g. starvation, rapamycin) to strengthen this argument.

Figure 2D shows that FAM134B interacts with CK2 both in the absence and presence of Torin and independent of the kinase activity of CK2, while the interaction of FAM134C decreases upon Torin treatment. How can this be explained given the SILAC data showing CK2 mediated phosphorylation of FAM134s upon Torin treatment? How is the interaction of CK2 with FAM134s regulated? Why are these IP experiments done with BafA1 treatment?

Figure 3 nicely shows that FAM134s are ubiquitinated upon Torin-mediated induction of ERphagy (FAM134C also in the absence of Torin) and that this is reduced with the CK2 inhibitor. They should use the phospho-mutants (used in Figure 4) to directly demonstrate the link between CK2-mediated phosphorylation and ubiquitination.

A recent publication (PMID: 36044577) demonstrate that mTORC1 inhibition limits FAM134C phosphorylation by CK2 to promote ER-phagy. The authors should discuss their data in relation to these observations.

The authors should generate FAM134B and/or FAM134B KO cells that are rescued with corresponding wild type or mutant constructs to be able to analyze the cellular/physiological implications of CK2-mediated ERphagy upon starvation and/or Torin treatment (e.g. ER stress, protein folding, metabolism).

Several of the experiments are only done once or twice. The authors should repeat all experiments at least 3 times and quantify their data.

Minor comments:

Line 127-129: this can be deleted as ERphagy already introduced

Line 161-164; sentence should be revised as Torin treatment do not induce mTOR-mediated ERphagy (rather mTOR inhibition-mediated).

Line 562-565: not so clear why there is a description of MEFs in this section.

Reviewer #3:

Remarks to the Author:

In the manuscript 'Function of ERphagy receptors is regulated via phosphorylation-dependent ubiquitination pathways' Berkane et al. find that the activity of FAM134B and FAM134C can be modulated by inhibition of mTORC1. They identified CK2 as the downstream kinase for Torin1-induced and FAM134-dependent ERphagy. The authors suggest that phosphorylation-dependent ubiquitination of both proteins might be the driving force behind FAM134B and FAM134C clustering. At this end, they leverage the quantitative super-resolution technic DNA-PAINT to determine the cluster size and density of FAM134B and FAM134C for DMSO, Torin1, and Torin1+CK2i treatment conditions.

Overall the study seems well-conceived, and the manuscript is mostly well-written. Especially the idea of leveraging quantitative super-resolution microscopy to determine the cluster size and density is clever and supports the punchline of the manuscript. I, however, have a couple of doubts about the execution of these experiments (see comments, suggestions, questions, and revisions).

In general, I think this manuscript could be a good fit for Nature Communications. However, I ask the authors to respond carefully to my comments, suggestions, questions, and revision wishes. In the case of an appropriate revision response, I would recommend the manuscript for publication in Nature Communications.

Comments, suggestions, questions, and revisions

Major revision points:

The density within the nanoscale cluster part of the paper is, in my opinion, problematic. The authors argue that the space between individual FAM134 proteins may vary. I am worried that this might also affect the antibody penetration. That would mean the fraction of labeled proteins would depend on the density of the cluster. Density analysis is difficult in that particular example.

- Have the authors considered doing the experiments with smaller labeling reagents (e.g., nanobodies)? One could, for example, tag the FAM134 protein with an ALFA-Tag and then target this ALFA-TAG with an ALFA-TAG Nanobody. These nanobodies are commercially available (see Nanotag or Massive Photonics).
- The authors use the P1 and the P5 sequence for the counting experiment. Both sequences have a k_{on} of roughly $1 \cdot 10^6 \text{ 1/(M} \cdot \text{sec)}$. The duration of the experiment is 2000 sec (40k frames at 20Hz and 20k frames at 10Hz). For the imager concentration, they use 400pM. For this concentration and the imager's association rate, it takes an average of 2500 sec for a docking site (i.e., the target of interest) to be sampled. That means not even half of the docking sites at the target of interest are sampled for the duration of 2000 sec of the author's experiment.
 - I, therefore, suggest redoing the experiments with an appropriate imaging duration. I suggest at least 5000 sec.
 - Alternatively, the authors could use the recently developed speed DNA-PAINT probes, allowing for up to 100x faster sampling. The duty cycle of these probes can be tuned into a similar range as for P1 and P5 so that there are minimal multiple binding events per diffraction-limited area.
- How reproducible are those measurements? Are the distributions reproducible if one does this measurement on three different days with three different samples? What is the sample-to-sample variation? I suggest presenting at least one independent repeat of these measurements in the supplementary information.

Minor revision points:

- Exchange-PAINT: The authors wrote several times in the manuscript that they did Exchange-PAINT (e.g., in the methods part and in the caption of figure 4). When I look at figure 4 and the main text, it is unclear what the authors did with Exchange-PAINT. There are neither two different channel images in the figure nor any further description in the main text about multiplexed imaging. From the methods section, it seems that the FAM proteins were imaged with an HA-TAG

antibody, and another protein was imaged with an LC3B antibody. Please explain better why you imaged that second protein and why it is not showing up in any figure.

- Since DNA-PAINT and Exchange-PAINT are essential elements of this manuscript, I suggest citing the original DNA-PAINT (DOI: 10.1021/nl103427w) paper and the original Exchange-PAINT (DOI: 10.1038/nmeth.2835) paper in addition to the already cited nature protocols paper.
- What is the y-axis in SI Figure 4A? Please label them.
- Is there a particular reason why the authors report the sigma of the cluster size without units? I suggest reporting it in nanometers.
- The average cluster size seems different for FAM134c compared to the mutant of FAM134c S258A. Do the authors have an idea why that is the case?
- It is not clear to me from either the results part or the figure if FAM134b under Torin1 treatment transitions to a bimodal distribution or if it is still another unimodal with a shifted mode.
 - Although it is stated in the discussion part of the paper that the distribution is bimodal, I think it is important to make this more clear in the results part and, even more importantly, in the figure, as this is a key result of the paper.
 - Line 276: 'Second cluster population' is this meant with respect to the DMSO treated case, or does it mean there is a second cluster population (i.e., bimodal) in the FAM134b Torin1 treated example?
 - I suggest changing Figure 4c. It is not so apparent what the Torin1 distribution (red) looks like since it overlaps with the other distributions. For example, one could use for all three distribution bars with just a stroke and no fill. Example: DOI '10.1016/j.bpr.2021.100032' Figure 2a-d.
- The authors state that they do 3D super-resolved imaging. What do they use the 3D information for? Have the authors considered filtering the values close to the minimum and the maximum axial detection range? If this filtering is not done, only partly detected clusters might bias the analysis.
- Line 312: 'The rationale behind this analysis is that an increase in protein density leads to a corresponding increase of single-molecule localizations detected per nanoscale cluster.' It would help the reader if the authors stated that the cluster size stays the same in that sentence.
- Line: 313: What do the authors want to say with this sentence? Is this supposed to mean: 'As expected, we detected an increased number of localizations with an increase in cluster diameter for all conditions?' If this is not what the authors meant, I wonder about an increase in respect to what?
- Figure 4 E - J: The authors state in the text that the triangles are black 316, but in the figure legend, they look colored. Please revise.
- Line 317/318: What is the rationale behind the thresholds (>100 loc/cluster, cluster diameter > 90 nm)?
- Line 269: 'Read-out' should be changed to readout for consistency
- Line 272: What does 'DNA-PAINT 2' mean?
- Line274: What does 'clustering algorithm 3' mean?
- Line386: It should be 'super-resolution' not 'super resolution'.
- Line419: (REF) ... I guess that meant the authors wanted to add a reference?

Reviewer #4:

Remarks to the Author:

The manuscript describes the analyses revealing that ERphagy is regulated by Ubiquitination of Fam134 proteins in a phosphorylation dependent fashion. This is nicely revealed using inhibitors, which point in the direction of the kinase CD2 being heavily involved in this process. Overall, a nice and thoroughly performed study which will contribute to the understanding of the ERphagy molecular mechanism.

Minor comments and concerns:

The authors write on page 6, line 133: "Since overexpressed FAM134C seemed to be relatively inactive under basal conditions, we hypothesize that mTOR-dependent phosphorylation events activate FAM134B/C-driven ERphagy." It seems like this hypothesis should have much more preliminary data for support? Please add these data if available or put in a reference.

The authors write on page 8 "several phosphorylation sites were dynamically regulated upon treatment (Fig. 2C)," However, there is no information on the statistics behind the quantitation of phosphosites, only an indication of identified phosphosites from the SILAC experiment? Much more information is needed for material and methods. Please include quantitation and statistics for each phosphosite in the manuscript. And please include all information on how the raw data was obtained, how many replicates the authors did, which database search program was used, which database was used, which parameters was used in the database searches and how was the quantitation and statistic performed.

Page 10 line 247-248: The sentence "High molecular species disappeared upon co-treatment of Torin1 and CK2 or treatment with E1 inhibitor..." should be "High molecular species disappeared upon co-treatment of Torin1 and CK2i or treatment with E1 inhibitor.."

We thank the reviewers for thoroughly reading, evaluating and commenting on our manuscript in a constructive manner, and foremost for their well-balanced overall positive comments. Here, we address to their comments in detail.

Our response to Reviewer #1

Pathways for regulation of induction of ER-phagy via cargo receptors are not well understood. Berkane et al propose that upon nutrient stress/mTOR inhibition that CK2 activity is in fact required to promote FAM134B phosphorylation, ubiquitination and clustering. FAM134C may be regulated more basally by this mechanism, although CK2 is clearly shown to be required for FAM134C induced ER-phagy. The pharmacologic screen to identify CK2 is very nice, but the authors' conclusions are somewhat preliminary/correlative and further data is in places required to confirm their proposal.

Q1) The authors should also cite and address the recent paper from Settembre lab (which has at least one author in common with this submission) that, superficially at least, seems to suggest the opposite conclusion, that CK2 in fact suppresses ER-phagy via FAM134C.

Response: We acknowledge the Reviewer's comment that our data seems to oppose findings published in di Lorenzo et al. Of note, our manuscript was submitted appx. 3 months before publication of Settembre's paper. This is why we did not comment on it in the first version of our manuscript.

In brief: The Settembre paper provides very interesting observations – upon others in-vivo, thereby stressing the physiological relevance of CK2-regulated phosphorylation of FAM134C. Here, they indeed suggest that phosphorylation by CK2 has a negative effect on ER-phagy. Based on an in-silico model they hypothesize that CK2-dependent phosphorylation of the FAM134C-LIR motif prevents ATG8 binding (1). However, actual in vitro data is not provided. In respect of the somehow contradictory results, we believe that there are additional regulators of the constitutive active kinase CK2 that allows cell/tissue-specific regulation of ER-phagy.

From our side, we now biochemically and biophysically analyzed the impact of FAM134C phosphorylation on its binding to ATG8 in vitro (**new Fig. S6**). More precisely, we expressed and purified FAM134C-LIR peptide from bacteria, phosphorylated it in vitro with purified CK2, and could show that phosphorylation of FAM134C-LIR by CK2 actually enhances ATG8 binding. In addition, synthetic phosphorylated and non-phosphorylated FAM134C-LIR peptides were analyzed in ITC measurements. In contrast to the hypothesis provided in the Settembre paper, phosphorylation led to an increase and not a decrease in ATG8 binding (= positive regulation of ER-phagy).

(New Fig. S6) **A**. ITC profiles for the titration of purchased FAM134-LIR (left) or FAM13C-pLIR (right) peptide into LC3B. **D,E**. Representative western blot image of GFP-Trap pulldown assay (**D**) and the relative bar plot (**E**) showing binding intensities of phosphorylated or non-phosphorylated GFP-FAM134C-LIR peptide (bait) to GST-LC3B (prey). GFP was used as a reference for ratio calculation. The statistical significance was estimated by one-way ANOVA. n=3, error bars=±STDV, [au]: arbitrary unit

We feel therefore confident to provide sufficient data to support an updated hypothesis for the findings published by the Settembre: a yet unknown regulatory factor with strong affinity to the phospho-LIR motif provides an additional level of regulation by masking the classical phospho-LIR of FAM134C thereby preventing binding to ATG8. Our results in combination with the in vivo data provided by the Settembre lab indicate a CK2-dependent mechanism by which organisms can achieve the necessary tissue-specific responses of FAM134-driven ER-phagy.

Q2) The CK2 data is supported entirely by use of pharmacologic inhibitors, which can be off-target. Does knockout of any of the key CK2s prevent FAM134B ubiquitylation/phosphorylation and/ or clustering?

Response: We acknowledge the Reviewer's valid concern about potential off targets of pharmacologic inhibitions in general. At the same time, we would like to emphasize that the CK2 inhibitor we used (SGC-CK2-1) is developed as a chemical probe (<https://www.thesgc.org/chemical-probes/SGC-CK2-1>), which has been intensively characterized and optimized. It is - unlike many other kinase inhibitors – highly selective.

CK2 is a constitutive active kinase whose activity is influenced by environmental circumstances and co-factors. Moreover, CK2 is an essential kinase and knockout of CK2 subunits is embryonic lethal (2). We therefore believe that knocking out CK2 in a cell line is an unfavorable experimental setup, because too many other pathways besides FAM134-driven ER-phagy would be long-term affected and intracellular communication altered. To assess the effect of CK2 depletion, we performed knockdown experiments using siPOOL targeting functional subunits of CK2 (CSNK2A1 and CSNK2A3) and analyzed FAM134B- and FAM134C-mediated ER-phagy (**new Fig. S3F,G**). Genetic depletion of CK2 reproduced the effect of our pharmacological inhibitor: CK2 depletion altered the capacity of FAM134 proteins to drive ER-phagy.

In addition, we used the same approach to evaluate the outcome of CK2 depletion on the ubiquitination status of FAM134 proteins (**new Fig. 4F,G**). We could here as well reproduce our previous data (**old Fig. 4C,E**) and show that CK2 depletion impacts negatively on mTOR inhibition-dependent ubiquitination of FAM134 proteins. Of note, during the revision time of this manuscript, a report published in nature (3) showed the relevance of ubiquitination in ER-phagy and highlighted its importance in regulating FAM134B-mediated ER-phagy and ER remodeling.

(New Fig. 4F,G) (F) Representative blot image and relative bar plot (G) of TUBE 2 assay in U2OS cells overexpressing FAM134B (left panel) or FAM134C (right panel) transfected with control siRNA (siCTRL) or siPOOL targeting CSNK2A1 and CSNK2A3 (siCK2). After 48h siRNA transfection, cells were DOX-induced overnight and treated for 6h with indicated conditions prior to cell lysis. anti-HA antibody was used to detect HA-tagged FAM134B and FAM134C and ubiquitinated species while anti-CK2 antibody was used to detect endogenous CK2 levels.

Overall, we feel confident to state that the effect we initially obtained from using chemical inhibition of CK2 results from a direct impact on CK2 activity and not of off-target effects.

Q3) The paper relies on overexpression of FAM134 paralogues throughout. It would be good to show that endogenous FAM134B/C clusters dependent upon CK2 activity, for example by imaging, and to show that endogenous FAM134B is ubiquitinated.

Response: we thank the Reviewer for pointing this out. We would like to emphasize that overexpression of FAM134 proteins is important to specifically monitor FAM134-mediated ER-phagy and minimize off-target effects caused by the presence of other endogenous ER-phagy receptors (ATL3, CCPG1, RTN3L, SEC62). It ensures that the presented phenotypes are on-target (FAM134-dependent) and provide a sufficient screening window.

Regarding super resolution microscopy, endogenous labeling is highly dependent on the availability of highly clean/specific antibodies. For FAM134 proteins such antibodies are currently not available. However, we ensured a reliable phenotype evaluation using ALFA-nanobody system at minimal expression level (please refer to answers to Reviewer 3 for more details about the system).

In brief this tool allows a precise assessment of protein assemblies through better accessibility of ALFA-tag nanobody into dense clusters due to its nano-scale size. We further optimized the experimental conditions to guarantee as low expression as possible of ALFA-tagged FAM134 proteins to approach the endogenous levels (0,1 μ g/ml Dox for 4h incubation time to express ALFA-FAM134B and no-DOX for leaky expression of ALFA-FAM134C). Our results presented in the **new Fig. S5E-F** confirm our previous observations and show that FAM134B and FAM134C clustering is mTOR inhibition- driven and that phosphorylation of FAM134 proteins by CK2 is a determinative step to ensure this process.

(New Fig. S5E) (E) Quantitative analysis of DNA-PAIN imaging data reports cluster diameters of ALFA-FAM134B (left panel) and ALFA-FAM134C (right panel) for DMSO, 250 nM Torin1 and 250 nM Torin1 + 1 μ M CK2 inhibitor. Cluster analysis was performed with the DBSCAN clustering algorithm in WT FAM134B- and FAM134C-overexpressing U2OS cells

Regarding endogenous ubiquitination of FAM134 proteins, we followed the Reviewer's advice and performed an endogenous pulldown of FAM134 proteins from cell lysates. **New Fig. 4C** shows the increase in FAM134B and FAM134C ubiquitination upon Torin1 treatment (ER-phagy induction) while co-treatment with CK2 inhibitor negatively impacted this ubiquitination on the endogenous level.

(New Fig. 4C) (C). Representative blot image of endogenously immunoprecipitated FAM134 proteins from total cellular extracts collected from U2OS cells previously treated for 6h with the indicated treatments. Anti-FAM134B or anti-FAM134C antibodies were used to detect endogenous proteins while anti-Ub (P4D1) was used to detect ubiquitination.

Q4) Phosphorylation sites are mapped in response to Torin via mass spectrometry but what is missing is direct demonstration that these are sensitive to CK2 inhibition in this analysis (I do acknowledge that some at least are a good fit for CK2 consensus sites). This is particularly important for the sites used as mutants later on to demonstrate the role of CK2-mediated phosphorylation in clustering. Does recombinant CK2 phosphorylated purified FAM134B at these sites?

Response: we thank the Reviewer for raising this key enquiry. We have made substantial efforts to provide a response to this question. We now are able to show that full length FAM134B purified from bacteria can be phosphorylated by purified CK2 *in vitro* (signal shift in a phos-tag gel) (new Fig.3C). Focusing on our identified phospho-sites, we purified FAM134B reticulon homology domain, phosphorylated *in vitro* by purified CK2 and subjected these samples for mass spectrometry. We could validate two out of three phospho-sites initially identified in the *in cellulo* mass spectrometry (new Fig. 3D and new Fig. S3D,E). We hypothesize the third phospho-site to be absent due to the non-optimal conditions (missing regulatory factors of CK2) in this experimental setup.

(New Fig. 3C,D) (C) Representative blot image of *in vitro* phosphorylation of full-length GST-FAM134B by purified CK2 and corresponding Coomassie-stained gel (lower panel). Anti-GST antibody was used to detect GST-FAM134B (upper panel). (D) Schematic three-dimensional representation of FAM134B-RHD highlighting serines 151 and 153 phosphorylated *in vitro* by CK2.

Q5) It's unclear what Figure 2D is supposed to be showing. It feels it is overlooked in the text.

Response : thanks for pointing this out, the figure was indeed not properly explained in the text. Old Fig.2D was supposed to show that CK2 is getting in close contact to FAM134 proteins, supporting the idea that it is able to directly target FAM134 proteins. With the new *in vitro* results of the purified system, it got obsolete and we removed it.

Q6) Fig. 3F - the majority of FAM134B/C IPd by anti-GFP, which I presume to indicate dimeric FAM134B/C, does not change upon Torin or CK2 inhibition. A minority Ub species does disappear with CK2 inhibition. So I understand that this indicates that a small amount of

multimeric FAM134B/C is ubiquitylated dependent upon CK2. However, this assay requires further controls. For instance, does the split-Venus interaction occur in cellulo or post-lysis? Without knowing this, it is difficult to say the ubiquitination occurs upon multimeric FAM134

Response: we acknowledge that we need to describe and explain this point/figure better. We actually do not want to claim that ubiquitination occurs upon multimerization of FAM134 (as

the Reviewer's comment indicate). On contrary, our combined data indicates that ubiquitination is a prerequisite for (extensive) multimerization/clustering and that phosphorylation is priming ubiquitination.

Stabilizing dimerization/multimerization of FAM134B using the Venus-split system, we do enrich the previously

reported, SDS-resistant high molecular species of FAM134B (Jiang et al., 2020). However, the ubiquitination pattern still depends on CK2 activity and therefore phospho-dependent ubiquitination. As such, multimerization of FAM134 proteins is insufficient to drive ubiquitination (old Fig. 3F now changed to Fig.4H).

Regarding the Reviewer's comment: A minority Ub species does disappear with CK2 inhibition. So I understand that this indicates that a small amount of multimeric FAM134B/C is ubiquitylated dependent upon CK2. The Reviewer might have been misled by the lower blot, which shows amounts of FAM134 oligomers (small differences). In the upper panel, which shows ubiquitination, the ubiquitination signal in the GFP-IP samples disappears completely upon CK2 inhibition (most right sample in the upper panel labeled with 'ubiquitinated FAM134B oligomers'; old Fig. 3F now changed to Fig.4H). We therefore show, that the ubiquitination of FAM134 is completely dependent on CK2.

Regarding the requested controls: split-Venus interactions do occur in cellulo and therefore pre-lysis as described in the original study characterizing this tool. This is indicated by the fluorescent GFP signal in intact cells. We now provide respective images as a control in the new Fig. S4E.

(New Fig. S4E) (E) Confocal fluorescence microscopy imaging of the BiCAP assay validating the *in cellulo* interaction between V1-FAM134B and V2-FAM134B or V1-FAM134C and V2-FAM134C. Fixed cells expressing the indicated plasmids were stained with REEP5 to mark the ER. Scale bar =10 μm

Q7) Figure 4/Supp Fig. 4 purport to show statistical analyses determining which cluster size distributions are probably different from one another, but I could not see the results of these analyses in the figure or the legend (e.g. p-val for Mann-Whitney U-test).

Response: the respective p-values are already added to the figures in the revised version.

MINOR POINTS

1) Preamble paragraph at start of Results chapter seems out of place

Response: we agree with the Reviewer and deleted this paragraph.

2) Missing (REF) on page 17

Response: we thank the Reviewer for spotting this. We added the missing reference.

3) Figure 1 - KDEL probe should be depicted within the ER lumen

Response: we thank the Reviewer for pointing this out. We modified the figure accordingly.

4) Abstract, "serin" typo, should be "serine"

Response: we apologize for the typo. We corrected this in the revised manuscript

5) page 8 - ATM/ATR/Chk1 are also all involved in sensing DNA damage, this should be acknowledged. Lane 202-204: For instance, ATM and ATR which orchestrate the DNA damage response (DDR) are members of the phosphatidylinositol 3-kinase PI3-related family (PI3KKs)

Response: we agree with the Reviewer and we now added a sentence to highlight the role of ATR/ATM/Chk1 in DNA damage.

Our response to Reviewer #2

In this manuscript, Berkane and co-workers demonstrate that the CK2 kinase phosphorylates the ER proteins FAM134B and FAM134C upon inhibition of mTOR (by Torin) and that this is required for their subsequent ubiquitination, clustering and function in ERphagy. CK2 was identified in a screen for drugs affecting FAM134B/C mediated ERphagy upon mTOR inhibition.

The manuscript is generally well written and the data are nicely presented. However, the data are largely descriptive and there is no link to any physiological relevance of their findings. To strengthen the manuscript the authors should address the following comments.

Specific comments:

Q1) The system used for the drug screen is based on cell lines with overexpression of both an ERphagy reporter (KDEL reporter) and HA-FLAG-FAM134B or HA-FLAG-FAM134C (FAM134C) that are treated with Torin to induce ERphagy. As overexpression of FAM134B/C in itself will induce ERphagy, this system might be somewhat artificial.

Response: We thank the Reviewer for raising this concern and we now added a better explanation of this point in the revised manuscript. Our data clearly shows that FAM134B/C overexpression does not cause extensive ER-phagy flux: we are measuring both basal (DMSO)

and Torin-induced ER-phagy flux in the presence of the same amount of overexpressed FAM134B/C. What FAM134B overexpression does cause is strong fragmentation of the ER (previously interpreted as ER-phagy induction), while fragments are not necessarily LC3B positive and/or in the

lysosome (4,5). A fragmentation phenotype is however not impacting / measured by the ER-phagy flux reporter used in this manuscript. Under basal (DMSO) conditions potential FAM134B-positive ER fragments are not delivered efficiently to the lysosome and therefore the GFP signal remains active (5) (this manuscript). Of note, overexpression of FAM134C does not even cause a fragmentation phenotype (5). As such, the presented effects on ER-phagy are not due to a disturbance of the ER network by FAM134 overexpression, but mirror endogenous behavior of the protein. The presented ER-phagy flux phenotypes upon Torin-induction and subsequent suppression of the phenotype by additional CK2 inhibition occurs independent of any overexpression phenotype.

To further strengthen these claims, we considered the Reviewers concern and investigated whether loss of endogenous Fam134b or Fam134c decreases the amount of Torin1-induced ER-phagy. (new Fig. 1C,D). For that, we used mouse embryonic fibroblasts (MEFs) isolated from *Fam134b* and *Fam134c* single knockout (KO) mice and complemented with the ssRFP-GFP-KDEL reporter (5) We could indeed confirm that the contribution of FAM134 proteins to mTOR-mediated ER-phagy is an endogenous function of FAM134 since their loss negatively impacts this process.

(New Fig. 1D) ER-phagy flux in ssRFP-GFP-KDEL *Fam134* WT and KO MEFs, represented by the RFP/GFP ratio of total integrated intensities in basal (DMSO) or induced (Torin1) conditions. n=3, error bars=±STDV. au= arbitrary unit

Q2) Although the authors do show that the CK2 inhibitor also has an effect on Torin-mediated ERphagy in control cells, it will be important to show directly that this is due to CK2-mediated phosphorylation of FAM134B/C. They need to demonstrate that CK2 does phosphorylate FAM134 and that expression of corresponding phospho-mutants are defect in ERphagy.

Response: we thank the Reviewer for raising this key enquiry. We have made substantial efforts to provide a response to this question. We now are able to show that full length FAM134B purified from bacteria can be phosphorylated by purified CK2 *in vitro* (signal shift in a phos-tag gel) (new Fig.3C). Focusing on our identified phospho-sites, we purified FAM134B reticulon homology domain, phosphorylated *in vitro* by purified CK2 and subjected these samples for mass spectrometry. We could validate two out of three phsopho-sites initially identified in the *in cellulo* mass spectrometry (new Fig. 3D and new Fig. S3D,E). We hypothesize the third phospho-site to be absent due to the non-optimal conditions (missing regulatory factors of CK2) in this experimental setup.

(New Fig. 3C,D) (C) Representative blot image of *in vitro* phosphorylation of full-length GST-FAM134B by purified CK2 and corresponding Coomassie-stained gel (lower panel). Anti-GST antibody was used to detect GST-FAM134B (upper panel). (D) Schematic three-dimensional representation of FAM134B-RHD highlighting serines 151 and 153 phosphorylated *in vitro* by CK2.

Regarding the second part, we followed the Reviewer's advice and investigated whether FAM134s phospho-mutants are defect in ER-phagy. Stable inducible cell lines expressing mutated FAM134B S149,151,153A or FAM134C S258A and ssRFP_GFP_KDEL in the background were created and subjected to the ER-phagy flux assay (**new Fig. 3E,F**). The FAM134C S258A mutation did not impact ER-phagy flux under basal conditions, however, had a strong negative impact on Torin1-induced ER-phagy flux. The FAM134B S149,151,153A mutant increased basal ER-phagy flux (we hypothesize stress upon dominant negative effects) – visible by a higher steady-state red/green ratio at 0h. As expected, the mutation of the FAM134B phosphorylation sites also negatively affected the ability of FAM134B to increase the ER-phagy flux after Torin1 treatment. We can therefore conclude that phosphorylation of FAM134 proteins at the RHD domain is functionally relevant for their ability to drive ER-phagy upon mTOR inhibition.

(**New Fig. 3E,F**) (**E**) Representative blot images validating the stable expression of FAM134 WT or phospho-mutant proteins under DOX induction in the background of U2OS cells expressing ER-phagy reporter ssRFP-GFP-KDEL. (**F**) ER-phagy flux in U2OS cells expressing FAM134 WT or phospho-mutants and ER-phagy reporter ssRFP-GFP-KDEL, represented by the RFP/GFP ratio of total integrated intensities in basal (DMSO) or induced (Torin1) conditions. n=3, error bars=±STDV

Q3) They should also demonstrate that knock-down of CK2 inhibits ERphagy to prove specificity of the inhibitor.

Response: we acknowledge the Reviewer's valid concern about the specificity of the inhibitor and potential off targets of pharmacologic inhibitions in general. To assess the effect of CK2 depletion, we performed knockdown experiments using siPOOL targeting functional subunits of CK2 (CSNK2A1 and CSNK2A3) and analyzed FAM134B- and FAM134C-mediated ER-phagy (**new Fig. S3F,G**). Genetic depletion reproduced the reported effect of pharmacological inhibition: CK2 depletion/inhibition alters the capacity of FAM134 proteins to drive ER-phagy.

(**New Fig. S3F,G**) (**F**) Representative blot image showing KD efficiency of CK2. (**G**) ER-phagy flux in ssRFP-GFP-KDEL U2OS cells overexpressing FAM134B or FAM134C with CK2 present or depleted in the background. ER-phagy flux was assessed either in basal (DMSO) or autophagy-induced (Torin1) conditions. Data points represent the averaged red/green ratio of three images comprising >150 cells and taken from three independent wells. n=3, error bars=±STDV, [au]: arbitrary unit.

Q4) The authors claim that CK2 mediated regulation of FAM134s and ERphagy is regulated downstream of mTOR. They should use other mTOR modulators (e.g. starvation, rapamycin) to strengthen this argument.

Response: we thank the Reviewer for this comment. We did repeat the experiment and tested several concentrations of Rapamycin and found that from 40 nM concentration, we could already see a similar yet weaker effect. Of note, Rapamycin only targets mTOR1 while Torin1 targets mTOR1 and mTOR2. We could also show that CK2 inhibition impacts the ability of FAM134B and FAM134C in driving ER-phagy triggered by Rapamycin. These findings support the robustness of our assay and the choice of Torin1 as a primary trigger to drive FAM134 protein activity (**new Fig. S1A**).

A

(New Fig. S1A) (A) ER-phagy flux in ssRFP-GFP-KDEL U2OS cells overexpressing FAM134B or FAM134C in basal (DMSO) or autophagy-induced (40nM Rapamycin) conditions. Total integrated fluorescent intensities (RFP/red and GFP/green) were monitored in the IncuCyte[®] S3 over a time course of 48h. Data points represent the averaged red/green ratio of three images comprising >150 cells and taken from three independent wells. n=3, error bars=±STDV, [au]: arbitrary unit

Even though broadly applied in the autophagy field, we believe that starvation is not an optimal treatment to induce selective autophagy: while mTOR inhibition is indeed one of the downstream effects of starvation, this harsh treatment causes a complete reprogramming of a cell including the activation of many other signaling pathways besides mTOR alteration. The interpretation of CK2 inhibition in such a setting would be highly variable.

Q5) Figure 2D shows that FAM134B interacts with CK2 both in the absence and presence of Torin and independent of the kinase activity of CK2, while the interaction of FAM134C decreases upon Torin treatment. How can this be explained given the SILAC data showing CK2 mediated phosphorylation of FAM134s upon Torin treatment? How is the interaction of CK2 with FAM134s regulated? Why are these IP experiments done with BafA1 treatment? Old Fig.2D was supposed to show that CK2 is getting in close contact to FAM134 proteins, supporting the idea that it is able to directly target FAM134 proteins. With the newly provided results of direct phosphorylation the purified system, old Fig2D got obsolete and we removed it.

Of note, upon inhibition a kinase may remain and stick to the (inactive) complex. We currently do not know which factors regulate CK2-dependent phosphorylation events. To unravel their identity would be highly interesting, however, exceeds the scope of this manuscript.

Q6) Figure 3 nicely shows that FAM134s are ubiquitinated upon Torin-mediated induction of ERphagy (FAM134C also in the absence of Torin) and that this is reduced with the CK2 inhibitor. They should use the phospho-mutants (used in Figure 4) to directly demonstrate the link between CK2-mediated phosphorylation and ubiquitination.

Response: As suggested by the Reviewer, we used the mentioned stable cell lines expressing mutated FAM134B S149,151,153A or FAM134C S258A and investigated their ubiquitination status when subjected to Torin1-mediated stress induction. FAM134B and FAM134C phospho-mutants are unmodified (no ubiquitination) both under basal and Torin1-induced conditions (**new Fig. S4C**).

(New Fig. S4C) Representative western blot showing enrichment for poly ubiquitinated FAM134 proteins using TUBE 2 assay. U2OS cells overexpressing FAM134B WT or S149-151-153A, FAM134C WT or S258A proteins were treated for 6 h with indicated conditions prior to cell lysis. TUBE2 assay was used to pulldown ubiquitinated entities. anti-HA antibody was used to detect HA-tagged FAM134B and FAM134C and ubiquitinated species.

Q7) A recent publication (PMID: 36044577) demonstrate that mTORC1 inhibition limits FAM134C phosphorylation by CK2 to promote ER-phagy. The authors should discuss their data in relation to these observations.

Response: Please refer to our answer to **Q1 of Reviewer 1**.

Q8) The authors should generate FAM134B and/or FAM134B KO cells that are rescued with corresponding wild type or mutant constructs to be able to analyze the cellular/physiological implications of CK2-mediated ERphagy upon starvation and/or Torin treatment (e.g. ER stress, protein folding, metabolism).

Response: we thank the Reviewer for the very interesting recommendation that helped us explore another axis in the functional relevance of FAM134s phosphorylation. As suggested, we reconstituted mouse embryonic fibroblasts (MEFs) lacking *Fam134b* (*Fam134b* KO) and *Fam134c* (*Fam134c* KO), respectively, with WT or phospho-mutant proteins FAM134B S149,151,153A or FAM134C S258A. We then used pro-Collagen I (COL1A1) as a readout to investigate the physiological significance of FAM134 phosphorylation. In contrast to the wild-type (WT) protein, FAM134 proteins lacking identified phospho-sites were unable to clear misfolded COL1A1.

(New Fig. 3G-J) **(G,H)** Representative blot image **(G)** and the relative bar plot **(H)** showing Collagen I accumulation in WT MEFs, *Fam134b* KO or cells reconstituted with FAM134B WT and phospho-mutant (S149-151-153A). **(I,J)** Representative blot image **(I)** and the relative bar plot **(J)** showing Collagen I accumulation in WT MEFs, *Fam134c* KO or cells reconstituted with FAM134C WT and phospho-mutant (S258A).

Q9) Several of the experiments are only done once or twice. The authors should repeat all experiments at least 3 times and quantify their data.

Response: we apologize for the lack of sufficient biological replicates in the first submission and now fixed this issue by repeating and quantified experiments where there was a gap.

Minor comments:

1) Line 127-129: this can be deleted as ERphagy already introduced
Response: we fixed this issue in the revised version of the manuscript

2) Line 161-164; sentence should be revised as Torin treatment do not induce mTOR-mediated ERphagy (rather mTOR inhibition-mediated).
Response: thank you for pointing this out. We specified the treatment in the revised version of manuscript.

3) Line 562-565: not so clear why there is a description of MEFs in this section.
Response: this was indeed less clear in the initial manuscript. With the newly provided data using MEFs cells, the description in this section is now more relevant.

Our response to Reviewer #3

In the manuscript 'Function of ERphagy receptors is regulated via phosphorylation-dependent ubiquitination pathways' Berkane et al. find that the activity of FAM134B and FAM134C can be modulated by inhibition of mTORC1m. They identified CK2 as the downstream kinase for Torin1-induced and FAM134-dependent ERphagy. The authors suggest that phosphorylation-dependent ubiquitination of both proteins might be the driving force behind FAM134B and FAM134C clustering. At this end, they leverage the quantitative super-resolution technic DNA-PAINT to determine the cluster size and density of FAM134B and FAM134C for DMSO, Torin1, and Torin1+CK2i treatment conditions.

Overall the study seems well-conceived, and the manuscript is mostly well-written. Especially the idea of leveraging quantitative super-resolution microscopy to determine the cluster size and density is clever and supports the punchline of the manuscript. I, however, have a couple of doubts about the execution of these experiments (see comments, suggestions, questions, and revisions).

In general, I think this manuscript could be a good fit for Nature Communications. However, I ask the authors to respond carefully to my comments, suggestions, questions, and revision wishes. In the case of an appropriate revision response, I would recommend the manuscript for publication in Nature Communications.

Comments, suggestions, questions, and revisions

Major revision points:

Q1) The density within the nanoscale cluster part of the paper is, in my opinion, problematic. The authors argue that the space between individual FAM134 proteins may vary. I am worried that this might also affect the antibody penetration. That would mean the fraction of labeled proteins would depend on the density of the cluster. Density analysis is difficult in that particular example.

- Have the authors considered doing the experiments with smaller labeling reagents (e.g., nanobodies)? One could, for example, tag the FAM134 protein with an ALFA-Tag and then

target this ALFA-TAG with an ALFA-TAG Nanobody. These nanobodies are commercially available (see Nanotag or Massive Photonics).

Response: We thank the Reviewer for pointing out this important consideration. We followed the suggestion of the Reviewer, and generated new stable inducible cell lines with FAM134B or FAM134C tagged with the small ALFA-tag (**new Fig. S5D**).

(**New Fig. S5D**) Super-resolved STED images showing DOX-induced expression (0.1 $\mu\text{g/ml}$) of ALFA-FAM134B (left panel) and ALFA-FAM134C (right panel) in U2OS cells. Fixed cells were stained with ALFA-tag nanobody to visualize FAM134B/C expression. Both images complement the results of SMLM, where FAM134B is sparsely populated in the ER and FAM134C is densely populated in the ER

We performed DNA-PAINT imaging using ALFA-TAG nanobodies (Massive-TAG-Q Anti-ALFA Custom, Massive Photonics, GmbH) coupled to a custom designed DNA-PAINT docking strand (5xR1, 5'-TCCTCCTCCTCCTCCTCCT-3') (**new Fig. S5E-I**). Our results fully support the initial results obtained using the HA-tag and anti-HA-tag antibodies.

(**New Fig. S5E-I**) (E) Quantitative analysis of DNA-PAINT imaging data reports cluster diameters of ALFA-FAM134B (left panel) and ALFA-FAM134C (right panel) for DMSO, 250 nM Torin1 and 250 nM Torin1 + 1 μM CK2 inhibitor. Cluster analysis was performed with the DBSCAN clustering algorithm in WT FAM134B- and FAM134C-overexpressing U2OS cells. (F-I) Comparative analysis of nanoscale clusters densities in ALFA-FAM134B (F, G) or ALFA-FAM134C (H, I). Fig. S5E-H represent an exact

replicate of the experiment conducted in Fig. 5 using ALFA-tag system and show reproducibility in these settings.

Q2) The authors use the P1 and the P5 sequence for the counting experiment. Both sequences have a k_{on} of roughly $1 \cdot 10^6 \text{ 1/(M} \cdot \text{sec)}$. The duration of the experiment is 2000 sec (40k frames at 20Hz and 20k frames at 10Hz). For the imager concentration, they use 400pM. For

this concentration and the imager's association rate, it takes an average of 2500 sec for a docking site (i.e., the target of interest) to be sampled. That means not even half of the docking sites at the target of interest are sampled for the duration of 2000 sec of the author's experiment.

◦ I, therefore, suggest redoing the experiments with an appropriate imaging duration. I suggest at least 5000 sec.

◦ Alternatively, the authors could use the recently developed speed DNA-PAINT probes, allowing for up to 100x faster sampling. The duty cycle of these probes can be tuned into a similar range as for P1 and P5 so that there are minimal multiple binding events per diffraction-limited area.

Response: We thank the Reviewer for this important suggestion regarding the kinetic parameters of DNA-PAINT. We followed the advice and performed the requested experiments that are now included in the revised manuscript. In order to ensure frequent and sufficient binding events of individual FAM134B and FAM134C proteins in DNA-PAINT, we performed additional experiments in new stable inducible cell lines expressing FAM134B-ALFA or FAM134C-ALFA, labeled with an ALFA-TAG nanobody carrying the 5xR1 docking strand (5'-TCCTCCTCCTCCTCCTCCT-3') (Massive-TAG-Q Anti-ALFA Custom, Massive Photonics, GmbH). We imaged 25.000 consecutive frames at 10 Hz using a ATTO655-labeled R1-imager strand (5'-AGGAGGA-3'). Assuming a k_{on} of about $37 \cdot 10^6$ (Ms)⁻¹ (<https://doi.org/10.1038/s41592-020-0869-x>), and considering an acquisition time of 2500 s and an imager strand concentration of 0.4 nM, we expect around 37 binding events per site; this expectation is matched in the new data now added to the manuscript.

Q3) How reproducible are those measurements? Are the distributions reproducible if one does this measurement on three different days with three different samples? What is the sample-to-sample variation? I suggest presenting at least one independent repeat of these measurements in the supplementary information.

Response We added the new measurements to the **new Fig. S5E-I**, which contain multiple measurements on at least 3 days for every condition. Additionally, the recently published article by González et al, reported that ubiquitination of FAM134B promotes receptor clustering under stress conditions applied by Torin1 treatment (3). Although different reports and experimenters, the communicated results seem to be in agreement with our data which suggests the high reproducibility of the performed measurements at different time points and in different labs.

Figure extracted from González et al : Relative frequency distribution of HA-FAM134B WT and HA-FAM134B 17KR cluster areas

Minor revision points:

- Exchange-PAINT: The authors wrote several times in the manuscript that they did Exchange-PAINT (e.g., in the methods part and in the caption of figure 4). When I look at figure 4 and the main text, it is unclear what the authors did with Exchange-PAINT. There are neither two

different channel images in the figure nor any further description in the main text about multiplexed imaging. From the methods section, it seems that the FAM proteins were imaged with an HA-TAG antibody, and another protein was imaged with an LC3B antibody. Please explain better why you imaged that second protein and why it is not showing up in any figure.

Response: we thank the Reviewer for pointing out this mistake. All our single-molecule localization measurements were made with single target, either FAM134B or FAM134C and we used only DNA-PAINT/speed-optimized DNA-PAINT. We now have updated the manuscript to account for this.

- Since DNA-PAINT and Exchange-PAINT are essential elements of this manuscript, I suggest citing the original DNA-PAINT (DOI: 10.1021/nl103427w) paper and the original Exchange-PAINT (DOI: 10.1038/nmeth.2835) paper in addition to the already cited nature protocols paper.

Response: We have revised the manuscript by adding the necessary references, one for the original DNA-PAINT paper (DOI: 10.1021/nl103427w) and one for the speed-optimized DNA-PAINT paper (<https://doi.org/10.1038/s41592-020-0869-x>).

- What is the y-axis in SI Figure 4A? Please label them.

Response: We have added the y-axis to the figure, which has now been updated to the **new supplementary figure 5A**.

- Is there a particular reason why the authors report the sigma of the cluster size without units? I suggest reporting it in nanometers.

Response: We have now added the units to the cluster size and apologize for not doing it in the first place.

- The average cluster size seems different for FAM134c compared to the mutant of FAM134c S258A. Do the authors have an idea why that is the case?

While at the moment our knowledge about FAM134C is still sparse, our data show that FAM134C is ubiquitinated already under basal conditions. This ubiquitination is absent in the mutant and may account for the size changes.

- It is not clear to me from either the results part or the figure if FAM134b under Torin1 treatment transitions to a bimodal distribution or if it is still another unimodal with a shifted mode.

Response: We aimed to emphasize the existence of two populations for FAM134B clusters upon Torin1 treatment (current Figure 5C, bimodal). We believe that the emerging, larger population (second population) represents active ER-phagy sites and tried to explain this better in the revised manuscript.

Using the ALFA-TAG system we observe only one FAM134B population (larger clusters, active ER-phagy sites) upon Torin1 treatment (supplementary figure 5E). We suspect that altered induction time/overexpression have caused this difference: in experiments using HA-labeled FAM134B, a longer induction time with DOX was used. Therefore, presumably more FAM134 proteins/clusters were present per cell that exceeded the capacity of other factors essential for the formation of active ER-phagy sites. As such only part of all present FAM134B clusters could be activated at a given time, causing the bimodal distribution into active (larger) and inactive (smaller) FAM134B clusters. The main message of the experiment (Torin1 treatment = elevated ER-phagy flux = larger clusters = active ER-phagy sites) remains consistent between experimental setups.

◦ Although it is stated in the discussion part of the paper that the distribution is bimodal, I think it is important to make this more clear in the results part and, even more importantly, in the figure, as this is a key result of the paper.

Response: we have now revised the text to take this into account.

◦ Line 276: 'Second cluster population' is this meant with respect to the DMSO treated case, or does it mean there is a second cluster population (i.e., bimodal) in the FAM134b Torin1 treated example?

Response: the second cluster population was meant with respect to the FAM134B Torin1 treated case and the newly emerging population.

◦ I suggest changing Figure 4c. It is not so apparent what the Torin1 distribution (red) looks like since it overlaps with the other distributions. For example, one could use for all three distribution bars with just a stroke and no fill. Example: DOI '10.1016/j.bpr.2021.100032' Figure 2a-d.

Response: we thank the Reviewer for the suggestion. We have three conditions summarized in one figure, which is always challenging to visualize. After comparing and revisiting our presentation we still believe our current layout visualizes the data in the best possible way.

• The authors state that they do 3D super-resolved imaging. What do they use the 3D information for? Have the authors considered filtering the values close to the minimum and the maximum axial detection range? If this filtering is not done, only partly detected clusters might bias the analysis.

Response: we thank the Reviewer for pointing this out. We only show 2D data in the manuscript and the text has now been revised to account for it.

• Line 312: 'The rationale behind this analysis is that an increase in protein density leads to a corresponding increase of single-molecule localizations detected per nanoscale cluster.' It would help the reader if the authors stated that the cluster size stays the same in that sentence.

Response: we thank the Reviewer for the suggestion. The manuscript has been changed on this regard.

• Line: 313: What do the authors want to say with this sentence? Is this supposed to mean: 'As expected, we detected an increased number of localizations with an increase in cluster diameter for all conditions?' If this is not what the authors meant, I wonder about an increase in respect to what?

Response: we thank the Reviewer for pointing out this ambiguity of the text. The manuscript has been revised accordingly.

• Figure 4 E - J: The authors state in the text that the triangles are black 316, but in the figure legend, they look colored. Please revise.

Response: we thank the Reviewer for pointing out this inconsistency, the text has been revised accordingly.

• Line 317/318: What is the rationale behind the thresholds (>100 loc/cluster, cluster diameter > 90 nm)?

Response: > 90 cluster diameter (separation of two populations from Figure 4C), >100 localizations (maximum number of localizations for FAM134B mutant for all cluster diameters. We have used the mutant cell line as a reference to set the filtering parameter

- Line 269: 'Read-out' should be changed to readout for consistency

Response: the manuscript has been changed accordingly.

- Line 272: What does 'DNA-PAINT 2' mean?

Response: we thank the Reviewer for pointing it out. The number corresponded to the reference which was formatted wrong. The manuscript has been changed accordingly.

- Line274: What does 'clustering algorithm 3' mean?

Response: we thank the Reviewer for pointing it out. The number corresponded to the reference which was formatted wrong. The manuscript has been changed accordingly.

- Line386: It should be 'super-resolution' not 'super resolution'.

Response: The manuscript has been changed accordingly.

- Line419: (REF) ... I guess that meant the authors wanted to add a reference?

Response: thank you for spotting this mistake, the missing reference was added.

Reviewer #4 (Remarks to the Author):

The manuscript describes the analyses revealing that ERphagy is regulated by Ubiquitination of Fam134 proteins in a phosphorylation dependent fashion. This is nicely revealed using inhibitors, which point in the direction of the kinase CD2 being heavily involved in this process. Overall, a nice and thoroughly performed study which will contribute to the understanding of the ERphagy molecular mechanism.

Minor comments and concerns:

The authors write on page 6, line 133: “Since overexpressed FAM134C seemed to be relatively inactive under basal conditions, we hypothesize that mTOR-dependent phosphorylation events activate FAM134B/C-driven ERphagy.” It seems like this hypothesis should have much more preliminary data for support? Please add these data if available or put in a reference.

Response: We thank the Reviewer for noticing the missing reference. We reported in our previous paper (5) that FAM134C and FAM134A are in a relative inactive state and need a stimulus to be activated. We now added the missing reference.

The authors write on page 8 “several phosphorylation sites were dynamically regulated upon treatment (Fig. 2C),” However, there is no information on the statistics behind the quantitation of phosphosites, only an indication of identified phosphosites from the SILAC experiment? Much more information is needed for material and methods. Please include quantitation and statistics for each phosphosite in the manuscript. And please include all information on how the raw data was obtained, how many replicates the authors did, which database search program

was used, which database was used, which parameters was used in the database searches and how was the quantitation and statistic performed.

Response: We extensively revised our material and method part in respect to mass spectrometry experiments/data. Additional information is available in the supplementary data file and the PRIDE database.

Username: reviewer_pxd043004@ebi.ac.uk

Password: IHYSD7k2

Page 10 line 247-248: The sentence “High molecular species disappeared upon co-treatment of Torin1 and CK2 or treatment with E1 inhibitor...” should be “High molecular species disappeared upon co-treatment of Torin1 and CK2i or treatment with E1 inhibitor..”

Response: we thank the Reviewer for pointing out this mistake. The text has been revised accordingly.

References:

1. Di Lorenzo G, Iavarone F, Maddaluno M, Belén Plata-Gómez A, Aureli S, Paz Quezada Meza C, et al. Phosphorylation of FAM134C by CK2 controls starvation-induced ER-phagy [Internet]. Vol. 8, Sci. Adv. 2022. Available from: <https://www.science.org>
2. Lou DY, Dominguez I, Toselli P, Landesman-Bollag E, O'Brien C, Seldin DC. The Alpha Catalytic Subunit of Protein Kinase CK2 Is Required for Mouse Embryonic Development. Mol Cell Biol. 2008 Jan 1;28(1):131–9.
3. González A, Covarrubias-Pinto A, Bhaskara RM, Glogger M, Kuncha SK, Xavier A, et al. Ubiquitination regulates ER-phagy and remodelling of endoplasmic reticulum. Nature [Internet]. 2023 Jun 8;618(7964):394–401. Available from: <https://www.nature.com/articles/s41586-023-06089-2>
4. Khaminets A, Heinrich T, Mari M, Grumati P, Huebner AK, Akutsu M, et al. Regulation of endoplasmic reticulum turnover by selective autophagy. Nature. 2015 Jun 18;522(7556):354–8.
5. Reggio A, Buonomo V, Berkane R, Bhaskara RM, Tellechea M, Peluso I, et al. Role of FAM134 paralogues in endoplasmic reticulum remodeling, ER-phagy, and Collagen quality control. EMBO Rep. 2021 Sep 6;22(9).

Reviewers' Comments:

Reviewer #1:

Remarks to the Author:

The authors have addressed all of my comments satisfactorily.

Reviewer #2:

Remarks to the Author:

The authors have done a very nice job addressing my comments and concerns and the manuscript is now acceptable for publication.

Reviewer #3:

Remarks to the Author:

The authors have addressed all my questions, suggestions, and comments in great detail. I am impressed with the revised version and I am recommending the manuscript for publication in Nature Communications.

Reviewer #4:

Remarks to the Author:

Comments to the annotated spectra:

FAM134B – S149/151/153:

The annotations for the peptide carrying S149 or 151 looks very similar and the b-ions in the low mass region is misplaced according to the masses provided in the figures. For example, for 149 the b3 ion should be 368 Da but is placed after 400 m/z... also, the majority of ions in the two spectra are just placed without any real peak.

Please provide annotated spectra for the low mass are so one can see the correct assigned ions that specifically localize the phosphosite. As they are now, one cannot judge the localization of the phosphate group and there is no evidence for several isoforms of the peptide.

The peptide identifying the S149/153 is a missed cleavage peptide at KP site. The authors used both trypsin and Lys-C. Trypsin does not cleave if there is a proline next to the lysine but Lys-C does.

Similar for the S153 peptide which is a full cleavage of the peptide that was used for the identification of S149/151, it is difficult to see the ions pinpointing the exact phosphosite. Please provide zoom-in figures. Also, it is strange that there are many peaks that are not assigned that are larger than the γ_{11} ion?

FAM134C – S258:

For the S258/260 annotated peptides there are the same problem. Here only b-ions can correctly assign the exact site for phosphorylation, and one cannot see the exact ions in the annotated spectra. The b-ions are also not linked to a specific peak in the spectra but spread out where the authors believe they should be. Please provide zoom-in of the lower mass area where the b-ions are so one can see if the b-ion peaks are there. Also, it is strange that one peptide is identified with oxidation and the other one is not? Normally oxidation of methionine is induced during sample preparation in proteomics and as such should only represent <20% of the ions of such a peptide.

FAM134C – S288/T283:

The fragment ion spectrum identifying the FAM134C – 288/T283 also can only be assigned using b-ions in the low mass area. Since there is only a b3 and b13 ion (and a b13* which I don't know what is?) if the peptide is correct then the phosphosite could be any of the S/T in the sequence.

FAM134C – S313:

This spectrum is decorated with multiple ions that make it very difficult to see anything. Please only use γ -ion, b-ions and * as loss of the phosphate group. Since the authors also label almost the complete γ -ion series with -H₂O the phosphate group could sit on every single S/T in the sequence. But again, difficult to see as the spectrum is decorated with potential fragments. Please change.

FAM134C - S320:

Basically, same spectrum as for S313... please pinpoint exactly the ions showing it is S320 and not S313.

FAM134C - S360 - ok

FAM134C - S126 - ok

FAM134B In Vitro - S153 - ok

FAM134B In Vitro - S151/153 - very weak identification and annotation.

Response to Reviewer #1 (Remarks to the Author):

The authors have addressed all of my comments satisfactorily.

We thank the reviewer for his/her time and the positive feedback.

Response to Reviewer #2 (Remarks to the Author):

The authors have done a very nice job addressing my comments and concerns and the manuscript is now acceptable for publication.

We thank the reviewer for his/her time and the positive feedback.

Response to Reviewer #3 (Remarks to the Author):

The authors have addressed all my questions, suggestions, and comments in great detail. I am impressed with the revised version and I am recommending the manuscript for publication in Nature Communications

We thank the reviewer for his/her time and the positive and motivating feedback.

Response to Reviewer 4#

We thank the reviewer for taking the time to review our manuscript and have a close look on our mass spectrometry data in particular. It certainly helped us to improve quality of the presentation and to be more precise regarding the specific sites of phosphorylation within the text. As suggested by the reviewer, we have in addition integrated the revised version of the spectra in our supplementary source data and mentioned their availability in the material and methods part. We now paid particular attention to possible displacement of annotated peaks as a result of the automated spectra representation of Maxquant andromeda. When needed, we provided zoomed-in graphs for better visualization and acknowledged the ambiguity of specific spectra in the text. A point-by-point response can be found below. We hope that the reviewer can now agree with our presentation and interpretation (text) of obtained data.

At this point, we would also like to stress that the mass spectrometry data presented in this manuscript is in principle not essential to validate our other findings. All reported phospho-sites on FAM134C as well as S151 of FAM134B have been found as well in (high-throughput screening) experiments performed in other labs and are reported in respective data bases (e.g. www.phosphosite.org), further supporting their existence and allowing to pick and concentrate on respective residues within a protein of interest for downstream analysis. We however prefer to base our studies on data confirmed within our lab whenever possible.

- 1) FAM134B – S149/151/153:

The annotations for the peptide carrying S149 or 151 looks very similar and the b-ions in the low mass region is misplaced according to the masses provided in the figures. For example, for 149 the b3 ion should be 368 Da but is placed after 400 m/z... also, the majority of ions in the two spectra are just placed without any real peak. Please provide annotated spectra for the low mass are so one can see the correct assigned ions that specifically localize the phosphosite. As they are now, one cannot judge the localization of the phosphate group and there is no evidence for several isoforms of the peptide.

The peptide identifying the S149/153 is a missed cleavage peptide at KP site. The authors used both trypsin and Lys-C. Trypsin does not cleave if there is a proline next to the lysine but Lys-C does.

Similar for the S153 peptide which is a full cleavage of the peptide that was used for the identification of S149/151, it is difficult to see the ions pinpointing the exact phosphosite. Please provide zoom-in figures. Also, it is strange that there are many peaks that are not assigned that are larger than the y11 ion?

To improve visualization, we now provide zoomed-in graphs.

Regarding the potentially missed cleavage, we agree with the reviewer on the presence of a proline in position +1 of the Lysine missed and that there is an additional missed Arginine at position 164. Independent of the cleavage specificity of Trypsin and LysC, it is known that phosphorylation can induce missed cleavages nearby the modified site (van der Laarse *et al.*, FEBS, 2019 and citations therein). The fact that we could identify the pS153 only as a fully tryptic peptide may indicate that phosphorylation of this serine does not induce a decreased digestion efficiency at this KP-cleavage site. Moreover, we identified the peptides reproducibly in several samples showing a global missed cleavage rate below 10%. While this digestion efficiency is quite decent, there still remains a certain rate of missed-cleaved peptides.

Unannotated peaks larger than y11 may be fragments originating from a peptide which was co-isolated by the quadrupole. However, such contamination does not invalidate the annotation and localization of the PTM.

We agree with the reviewer on the ambiguity of the localization and therefore rephrased our finding to more accurately depict the phosphorylation of the peptide: Fig. 3B reports S149/S151/S153 now as a single finding and we edited the text accordingly. In other words, the peptide containing S149/151/153 might be phosphorylated at any of the three locations, yet only one of them seems to be phosphorylated in the detected peptides. CK2 is known for promiscuous phosphorylation of clustered serines, yet we could not detect any multiply-phosphorylated isoforms. Additionally, the coisolation of the different mono-phosphorylated isoforms also biases the relative ion abundance in the resulting MS/MS-spectra towards fragments without the pS/T, which makes it even harder to detect the specific fragments in standard proteomics experiments as performed in this study. We still report all of the spectra from individual serines as we believe it is valuable for the field, as ambiguity does not prove or disprove the finding.

2) FAM134C – S258:

For the S258/260 annotated peptides there are the same problem. Here only b-ions can correctly assign the exact site for phosphorylation, and one cannot see the exact

ions in the annotated spectra. The b-ions are also not linked to a specific peak in the spectra but spread out where the authors believe they should be. Please provide zoom-in of the lower mass area where the b-ions are so one can see if the b-ion peaks are there. Also, it is strange that one peptide is identified with oxidation and the other one is not? Normally oxidation of methionine is induced during sample preparation in proteomics and as such should only represent <20% of the ions of such a peptide.

We agree with the reviewer on the ambiguity of the localization, and therefore rephrased throughout the text our finding to more accurately depict the phosphorylation of the peptide: In Fig. 3B S258/260 are now reported as a single finding.

We agree with the reviewer that the oxidation found in one of the peptides containing our desired modification is present. Oxidation of methionine arises from sample preparation, which in this particular sample was non-optimal. At the same time, we would like to stress that although the MS/MS spectra with the highest localization probability used for the figures came from an oxidized peptide, we did also identify the non-oxidized, but phosphorylated isoform (see table below). As such, we have found the phosphorylation in two independent peptides and for that reason we think that the oxidation does not invalidate the finding and actually is strengthening the Identification.

Sequence	Modifications	Mass	Gene Names	Unique (Groups)	Missed cleavages	PEP	MS/MS scan number	Score	Delta score	Intensity	Best MS/MS	MS/MS Count
AMDNHSDSEELAAFCPQLDDSTVAR	Oxidation (M);Phospho (STY)	3003,19	FAM134C	yes	0	1.70E-20	28393	186,86	179,91	2435000000	13514	34
AMDNHSDSEELAAFCPQLDDSTVAR	Phospho (STY)	2,987,195	FAM134C	yes	0	3.16E-10	28611	148,89	140,68	1085600000	13546	15

3) FAM134C – S288/T283:

The fragment ion spectrum identifying the FAM134C – 288/T283 also can only be assigned using b-ions in the low mass area. Since there is only a b3 and b13 ion (and a b13* which I don't know what is?) if the peptide is correct then the phosphosite could be any of the S/T in the sequence.

We agree with the reviewer on the ambiguity of the localization, and therefore rephrased throughout the text our finding to more accurately depict the phosphorylation of the peptide: T283/S285/288 are now reported as a single finding.

The ion b13* refers to the ion with a phosphate loss.

4) FAM134C – S313:

This spectrum is decorated with multiple ions that make it very difficult to see anything. Please only use y-ion, b-ions and * as loss of the phosphate group. Since the authors also label almost the complete y-ion series with -H2O the phosphate group could sit on every single S/T in the sequence. But again, difficult to see as the spectrum is decorated with potential fragments. Please change.

We have cleaned and trimmed the spectrum for better representation.

5) FAM134C – S320:

Basically, same spectrum as for S313... please pinpoint exactly the ions showing it is S320 and not S313.

We have cleaned and trimmed the spectrum for better representation. While the spectra are very similar, especially the y-ion series pinpoints the phosphorylation to either S320 (y9 -y22), or S313 (y16 -y22). As stated above, the co-isolation of the peptide isoforms makes it hard to detect differentiating ions bearing the phosphorylation, also for the search engine. In this case, one can clearly see evidence for both isoforms in the two spectra, also because there are two prolines at both ends of the peptide. The Proline-effect leads to increased fragment intensities at these bonds and we can see them very well for the two isoforms, further strengthening the identification of the two mono-phosphorylated isoforms.

6) FAM134C – S360 – ok

7) FAM134C – S126 – ok

8) FAM134B In Vitro – S153 – ok

9) FAM134B In Vitro – S151/153 – very weak identification and annotation.

This experiment was exclusively performed to prove that CK2 is in principle able to directly phosphorylate FAM134B. The spectra of FAM134B-S153 confirms this. We therefore removed the - admittedly very weak – spectrum for S151/153 from the reported data set and changed the text accordingly.

FAM134B-S149

Raw File: 20220311_MIA_LPS_FAM134B_DMSO_4
 Scan: 20589
 Method: FTMS; HCD
 Score: 147.67
 m/z: 737.35
 Gene names: FAM134B

FAM134B-S149 Zoom in lower ions

Raw file: 20220311_LMAA_PS_FAM134B_DMSO_4
 Scan #: 20569
 Method: FTMS_TICD
 Score: 147.67
 m/z: 737.35
 Gene names: FAM134B

Y17* L S E S W E V I N S K P D E R P -
 b2* b3* b4* b5* b6* b7* b8* b9* b10* b11*
 Y4 Y5 Y6 Y7 Y8 Y9 Y10 Y11 Y12 Y13 Y14 Y15 Y16

FAM134B-S151

FAM134B-S151 Zoom in lower ions

FAM134B-S153

Raw file: 20220311_MAA_PS_FAM6_DMSO_4
 Scan: 23731
 Method: FTMS; FCD
 Score: 11925
 m/z: 728.63
 Gene names: FAM134B

FAM134B-S153 Zoom in lower ions

FAM134B-S153 Zoom in Y8*

FAM134C-S258

Row File: 20220314_PP_F5_FAM134C_DW50_4
 Score: 186,86
 Method: FTMS, HCD
 Gene names: FAM134C

- A **M** **D** **N** **H** **S** **D** **S** **E** **E** **L** **A** **A** **F** **C** **P** **Q** **L** **D** **D** **S** **T** **V** **A** **R**
 b5 b7 b8 b9 b10 b11 b12 b13 b14 b15
 Y14 Y13 Y12 Y11 Y10 Y9 Y8 Y7 Y6 Y5 Y4 Y3 Y2 Y1

FAM134C-S258- Low ions

Raw File: 20220314_HR_P5_FAM134C_DMSO_4

Scan: 28393

Method: FTMS; HCD

Score: 186.86

m/z: 1002.41

Gene names: FAM134C

- A M D N H S D S E E L A A F C P Q L D D S T V A R

b2
b5
b7
b8
b9
b10
b11
b12
b13
b14
b15
Y1
Y2
Y3
Y4
Y5
Y6
Y7
Y8
Y9
Y10
Y11
Y12
Y13
Y14

FAM134C-S258- High ions

Raw File: 20220314_RR_PS_FAM134C_DMSO_4 Scan: 20393 Method: FTMS, 11CD Score: 18686 m/z: 1062.41 Gene Names: FAM134C

- A M D N H S D S E E L A A F C P Q L D D S T V A R

b2 b5 b7 b8 b9 b10 b11 b12 b13 b14 b15
 Y1 Y2 Y3 Y4 Y5 Y6 Y7 Y8 Y9 Y10 Y11 Y12 Y13 Y14 Y15

FAM134C-S260

Raw File: 20220314_BR_P5_FAM134C_DMSO_3
 Scan: 28119
 Method: FAMS-MSD
 Score: 146.27
 m/z: 707
 Gene names: FAM134C

FAM134C-S260 – Low ions

Raw file: 20220914_BR_P5_FAM134C_DMSO_3

Scan: 28119

Method: FTMS_TICD

Score: 146327

m/z: 399.07

Gene names: FAM134C

- A M D N H S D S E E E L A A F C P Q L D D S T V A R
 [a2] [b5] [b7] [b8] [b9] [b10] [b11] [b12] [b13] [b14] [b15] [b16] [b17] [b18] [b19] [b20] [b21] [b22]

FAM134C-S260 – High ions

FAM134C-T283

Raw file: 20220314_LRR_PS_FAM134C_T0283.L
 Scan: 24631
 Method: FIMS, HCD
 Score: 64.65
 m/z: 303.4
 Gene names: FAM134C

FAM134C-T283 – Low Ions

FAM134C-T283 – High Ions

Raw file: 20220314_PP_F5_FAM134C_T08B.L1
 Scan: 24601
 Method: FTMS, HCD
 Score: 64.65
 m/z: 303.4
 Gene names: FAM134C

E L A I P D S E H S D A E V S C T D N G T F N L S R -
 b2 b3 b4 b12 b13

FAM134C-S285

Raw File: 20220314_RIV_P5_FAM134C_DMSD_4
 Scan: 24933
 Method: FTMS; HCD
 Score: 196.78
 m/z: 983.75
 Gene names: FAM134C

FAM134C-S285- Low Ions

Raw File: 20220314_RR_PS_FAMC_DMSO_4

SMS: 24933

Method: FTMS_HCD

Scan: 19678

RT: 583.75

Event: FAM134C

FAM134C-S288

FAM134C-S313

FAM134C-S313- Low ions

FAM134C-S313- Mid ions

FAM134C-S313- High ions

FAM134C-S320

FAM134C-S320 – low Ions

Raw File: 20220318_BR_PS_FAMC_Tomu_ATR_2

Scan: 22081

Method: FIMS_HCD

Score: 493.6

m/z: 606

Gene names: FAM134C

FAM134C-S320 – Mid Ions

Raw File: 20220318_PP_05_FAM1C_Term_ATR_2
 Scan: 22881
 Method: FIMS;TICD
 Score: 493.6
 m/z: 1326.06
 Gene names: FAM134C

FAM134C-S320 – High Ions

FAM134C-S360 – High Ions

In Vitro FAM134B-S153

Raw File: 20230329_JHF_LCL_MHO_PS_026_FAM134B_RHD_CK1_01
Scan: 19333 Method: FTMS;HCD Score: 100.04 m/z: 737.02

- S L S E W E V I N S K P D E R R -
[b1] [b2] [b3] [b4] [b5]

Reviewers' Comments:

Reviewer #4:

Remarks to the Author:

Ok